# CodeGen: An Open Large Language Model for Code with Multi-Turn Program Synthesis

**Erik Nijkamp**[*], **Bo Pang**[*], **Hiroaki Hayashi**[*],
**Lifu Tu**, **Huan Wang**, **Yingbo Zhou**, **Silvio Savarese**, **Caiming Xiong**

Salesforce Research

## Abstract

Program synthesis strives to generate a computer program as a solution to a given problem specification, expressed with input-output examples or natural language descriptions. The prevalence of large language models advances the state-of-the-art for program synthesis, though limited training resources and data impede open access to such models. To democratize this, we train and release a family of large language models up to 16.1B parameters, called CodeGen, on natural language and programming language data, and open source the training library JAXFORMER. We show the utility of the trained model by demonstrating that it is competitive with the previous state-of-the-art on zero-shot Python code generation on HumanEval. We further investigate the multi-step paradigm for program synthesis, where a single program is factorized into multiple prompts specifying subproblems. To this end, we construct an open benchmark, Multi-Turn Programming Benchmark (MTPB), consisting of 115 diverse problem sets that are factorized into multi-turn prompts. Our analysis on MTPB shows that the same intent provided to CodeGen in multi-turn fashion significantly improves program synthesis over that provided as a single turn. We make the training library JAXFORMER and model checkpoints available as open source contribution: `https://github.com/salesforce/CodeGen`.

## 1 Introduction

Creating a program has typically involved a human entering code by hand. The goal of program synthesis is to automate the coding process, and generate a computer program that satisfies the user's specified intent. Some have called it the holy grail of computer science (Manna & Waldinger, 1971; Gulwani et al., 2017). Successful program synthesis would not only improve the productivity of experienced programmers but also make programming accessible to a wider audience.

Two key challenges arise when striving to achieve program synthesis: (1) the intractability of the search space, and (2) the difficulty of properly specifying user intent. To maintain an expressive search space, one needs a large search space, which poses challenges in efficient search. Previous work (Joshi et al., 2002; Panchekha et al., 2015; Cheung et al., 2013) leverages domain-specific language to restrict the search space; however, this limits the applicability of synthesized programs. On the contrary, while being widely applicable, general-purpose programming languages (*e.g.*, C, Python) introduce an even larger search space for possible programs. To navigate through the enormous program space, we formulate the task as language modeling, learning a conditional distribution of the next token given preceding tokens and leverage transformers (Vaswani et al., 2017) and large-scale self-supervised pre-training. This approach has seen success across modalities (Devlin et al., 2019; Lewis et al., 2020; Dosovitskiy et al., 2021). Likewise, prior works have developed pre-trained language models for programming language understanding (Kanade et al., 2020; Feng et al., 2020).

To realize program synthesis successfully, users must employ some means to communicate their intent to the models such as a logical expression (which specifies a logical relation between inputs

---

[*] Equal contribution.

Correspondence to: Erik Nijkamp (erik.nijkamp@salesforce.com), Bo Pang (b.pang@salesforce.com), Hiroaki Hayashi (hiroakihayashi@salesforce.com), Yingbo Zhou (yingbo.zhou@salesforce.com), Caiming Xiong (cxiong@salesforce.com).

and outputs of a program), pseudo-code, input-output examples, or a verbalized specifications in natural language. On the one hand, a complete formal specification enjoys the exact specifications of user intent but may require domain expertise and effort from users to translate the intent to such a form. On the other hand, specification merely based on input-output examples is less costly but may under-specify the intent, leading to inaccurate solutions. Previous work has benefited from various methods and their combinations as the input to program synthesis models, including pseudo-code (Kulal et al., 2019), a part of a program and its documentation (Chen et al., 2021), or natural language paragraph with input-output examples (Hendrycks et al., 2021). However, we argue that a truly user-friendly form of intent is natural language text.

To overcome these challenges, we propose a multi-turn program synthesis approach, where a user communicates with the synthesis system by progressively providing specifications in natural language while receiving responses from the system in the form of synthesized subprograms, such that the user together with the system complete the program in multiple steps. The following two considerations motivate this approach.

First, we speculate that factorizing a potentially long and complicated specification into multiple steps would ease the understanding by a model and hence enhance program synthesis. In the multi-turn approach, a model can focus on the specification associated with one subprogram and avoid arduously tracking the complicated dependency among subprograms. This effectively reduces the search space besides the convenience of specifying user intent. Indeed, our speculations are confirmed in our experiments with higher quality synthesized programs through the multi-turn approach.

Second, code exhibits a weak pattern of interleaved natural and programming language, which may be exploitable. Such a pattern is formed by programmers who explain the functionality of a program with comments. With the language modeling objective, we hypothesize that the interleaving pattern provides a supervision signal for the model to generate programs given natural language descriptions over *multiple* turns. The signal is highly noisy or weak, because only a subset of data would exhibit such a pattern, comments may be inaccurate or uninformative, and some of them may even be placed at an irrelevant position. However, up-scaling the model and data size might overcome such weak supervision, allowing the model to develop multi-turn program synthesis capacity. This enables user intent to be expressed in multiple turns, that is, the intent can be decomposed and fulfilled part by part while each turn can easily be expressed in natural language.

In this work, we develop a multi-turn programming benchmark to measure the models' capacity for multi-turn program synthesis. To solve a problem in the benchmark, a model needs to synthesize a program in multiple steps with a user who specifies the intent in each turn in natural language. Please refer to Figure 1 for an example where the model synthesizes a program to extract the user name of an email address. Performance on the benchmark is measured by pass rate on expert-written test cases. To the best of our knowledge, this is the first multi-turn program synthesis benchmark, which allows quantitative analysis of multi-turn program synthesis. With the emergence of multi-turn program synthesis capacity in large language models that benefits problem-solving, we believe this benchmark will foster future research in program synthesis.

**Our Contributions** Our work shares the basic idea of adopting language models for program synthesis with the recent and concurrent efforts (Chen et al., 2021; Austin et al., 2021; Li et al., 2022) with a single-turn user intent specification. In addition, we contribute with respect to four aspects:

- We study multi-turn program synthesis emerging in autoregressive models under scaling laws.
- We leverage this capacity to introduce a multi-turn program synthesis paradigm.
- We investigate its properties quantitatively with a novel multi-turn programming benchmark.[1]
- We will open source model checkpoints[2] and the custom training library: JAXFORMER.[3]

For program synthesis, no large-scale models competitive with Codex are available as open-source. This hinders progress, given that the expensive compute resources required to train these models are only accessible to a limited number of institutions. Our open source contribution allows a wide range of researchers to study and advance these models, which may greatly facilitate research progress.

---

[1]Benchmark: `https://github.com/salesforce/CodeGen/tree/main/benchmark`

[2]Checkpoints: `https://github.com/salesforce/CodeGen`

[3]Training: `https://github.com/salesforce/jaxformer`

## 2 MODEL TRAINING

To evaluate the emergence of multi-turn programming capabilities under scaling laws, we adopt standard transformer-based autoregressive language models, varying (1) the number of model parameters (350M, 2.7B, 6.1B, 16.1B) and (2) the number of tokens of programming languages in the training corpora. For scaling the training, a custom library JAXFORMER for TPU-v4 hardware was developed and will be released as open-source, including the trained model weights.

### 2.1 DATASETS

The family of CODEGEN models is trained sequentially on three datasets: THEPILE, BIGQUERY, and BIGPYTHON.

The natural language dataset THEPILE is an 825.18 GiB English text corpus collected by Gao et al. (2020) for language modeling (MIT license). The dataset is constructed from 22 diverse high-quality subsets, one of which is programming language data collected from GitHub repositories with >100 stars that constitute 7.6% of the dataset. Since the majority of THEPILE is English text, the resulting models are called as natural language CODEGEN models (CODEGEN-NL).

The multi-lingual dataset BIGQUERY is a subset of Google's publicly available BigQuery dataset, which consists of code (under open-source license) in multiple programming languages. For the multi-lingual training, the following 6 programming languages are chosen: C, C++, Go, Java, JavaScript, and Python. Thus, we refer to models trained on the BIGQUERY as multi-lingual CODEGEN models (CODEGEN-MULTI).

The mono-lingual dataset BIGPYTHON contains a large amount of data in the programming language, Python. We have compiled public, non-personal information from GitHub consisting of permissively licensed Python code in October 2021. Consequently, we refer to models trained on BIGPYTHON as mono-lingual CODEGEN models (CODEGEN-MONO).

The pre-processing follows: (1) filtering, (2) deduplication, (3) tokenization, (4) shuffling, and (5) concatenation. For details on THEPILE, we refer to Gao et al. (2020). For BIGQUERY and BIGPYTHON, we refer to Appendix A. Table 5 summarizes the statistics of the training corpora.

### 2.2 MODELS

The CODEGEN models are in the form of autoregressive transformers with next-token prediction language modeling as the learning objective trained on a natural language corpus and programming language data curated from GitHub. The models are trained in various sizes with 350M, 2.7B, 6.1B, and 16.1B parameters. The first three configurations allow for direct comparison with open-sourced large language models trained on text corpus, GPT-NEO (350M, 2.7B) (Black et al., 2021) and GPT-J (6B) (Wang & Komatsuzaki, 2021). See Table 6 in Appendix A for model specifications.

The CODEGEN models are trained in a sequential nature over datasets. CODEGEN-NL is first trained on THEPILE. CODEGEN-MULTI is initialized from CODEGEN-NL and trained on BIGQUERY. Finally CODEGEN-MONO is initialized from CODEGEN-MULTI and trained on BIGPYTHON.

The emergence of program synthesis conditional on descriptions in natural language may stem from the size of the models and data, training objective, and nature of the training data itself. This is called emergence since we do not explicitly train the model on comment-code pairs. Similar phenomena are observed in a wide range of natural language tasks where a large-scale unsupervised language model can solve unseen tasks in a zero-shot fashion (Brown et al., 2020). The emergence phenomena or surprising zero-shot generalization is often attributed to the large scale of the model and the data.

While our focus is not to reveal the underlying mechanism on why program synthesis capacity emerges from simple language modeling, we make an attempt to provide an explanation given the nature of our modeling approach and the training data. The data consists of regular code from GitHub (without manual selection), for which *some* data exhibits a pattern of interleaved natural and programming language, which we believe provides a noisy supervision signal for the program synthesis capacity due to the next-token prediction training objective. However, we emphasize that such a data pattern is highly noisy and weak, because only a subset of data exhibits such a pattern, e.g., comments may be inaccurate or uninformative, and some of them may even be placed at an irrelevant

| Model | pass@$k$ [%] | | |
|---|---|---|---|
| | $k = 1$ | $k = 10$ | $k = 100$ |
| GPT-NEO 350M | 0.85 | 2.55 | 5.95 |
| GPT-NEO 2.7B | 6.41 | 11.27 | 21.37 |
| GPT-J 6B | 11.62 | 15.74 | 27.74 |
| CODEX 300M | 13.17 | 20.37 | 36.27 |
| CODEX 2.5B | 21.36 | 35.42 | 59.50 |
| CODEX 12B | 28.81 | 46.81 | 72.31 |
| code-cushman-001* | 33.5 | 54.3 | 77.4 |
| code-davinci-001* | 39.0 | 60.6 | 84.1 |
| code-davinci-002* | 47.0 | 74.9 | 92.1 |
| CODEGEN-NL 350M | 2.12 | 4.10 | 7.38 |
| CODEGEN-NL 2.7B | 6.70 | 14.15 | 22.84 |
| CODEGEN-NL 6.1B | 10.43 | 18.36 | 29.85 |
| CODEGEN-NL 16.1B | 14.24 | 23.46 | 38.33 |
| CODEGEN-MULTI 350M | 6.67 | 10.61 | 16.84 |
| CODEGEN-MULTI 2.7B | 14.51 | 24.67 | 38.56 |
| CODEGEN-MULTI 6.1B | 18.16 | 28.71 | 44.85 |
| CODEGEN-MULTI 16.1B | 18.32 | 32.07 | 50.80 |
| CODEGEN-MONO 350M | 12.76 | 23.11 | 35.19 |
| CODEGEN-MONO 2.7B | 23.70 | 36.64 | 57.01 |
| CODEGEN-MONO 6.1B | 26.13 | 42.29 | 65.82 |
| CODEGEN-MONO 16.1B | **29.28** | **49.86** | **75.00** |

Table 1: Evaluation results on the HumanEval benchmark. Each pass@$k$ (where $k \in \{1, 10, 100\}$) for each model is computed with three sampling temperatures ($t \in \{0.2, 0.6, 0.8\}$) and the highest one among the three are displayed, which follows the evaluation procedure in Chen et al. (2021). Results for the model marked with * are from Chen et al. (2022).

position. Therefore, we believe two main factors contribute to the program synthesis capacity: 1) large scale of model size and data size and 2) noisy signal in training data.

The scaling of such LLMs requires data and model parallelism. To address these requirements, a training library JAXFORMER (https://github.com/salesforce/jaxformer) was developed for efficient training on Google's TPU-v4 hardware. We refer to Appendix A for further details on the technical implementation and sharding schemes. Table 6 summarizes the hyper-parameters.

## 3  SINGLE-TURN EVALUATION

We first evaluate our CODEGEN using an existing program synthesis benchmark: HumanEval (MIT license) (Chen et al., 2021). HumanEval contains 164 hand-written Python programming problems. Each problem provides a prompt with descriptions of the function to be generated, function signature, and example test cases in the form of assertions. The model needs to complete a function given the prompt such that it can pass all provided test cases, thus measuring the performance by functional correctness. Since a user intent is specified in a single prompt and provided to the model once, we regard the evaluation on HumanEval as a single-turn evaluation, to distinguish it from the multi-turn evaluation which we introduce in the next section. Following Chen et al. (2021), we recruit nucleus sampling (Holtzman et al., 2020) with top-$p$ where $p = 0.95$.

### 3.1  HUMANEVAL PERFORMANCE SCALES AS A FUNCTION OF MODEL SIZE AND DATA SIZE

We compare our models to the Codex models (Chen et al., 2021), which demonstrate the state-of-the-art performance on HumanEval. Moreover, our models are compared to open-sourced large language models, GPT-NEO (Black et al., 2021) and GPT-J (Wang & Komatsuzaki, 2021). These are trained on THEPILE (Gao et al., 2020), and thus similar to our CODEGEN-NL models, in terms of training data and model size. All models are evaluated with temperature $t \in \{0.2, 0.6, 0.8\}$, and we compute pass@$k$ where $k \in \{1, 10, 100\}$ for each model. For direct comparison to the results by Chen et al. (2021), we choose the temperature that yields the best-performing pass@$k$ for each

| CODEGEN-MONO | 350M | 2.7B | 6.1B | 16.1B |
|---|---|---|---|---|
| Pass | $3.78 \pm 0.23$ | $3.66 \pm 0.14$ | $3.35 \pm 0.13$ | $3.12 \pm 0.11$ |
| Non-Pass | $5.18 \pm 0.19$ | $4.37 \pm 0.18$ | $3.88 \pm 0.13$ | $3.40 \pm 0.11$ |

Table 2: Average prompt perplexity$^{\downarrow}$ ($\pm$ standard error) of CODEGEN-MONO models on pass and non-pass problems.

$k$. The results of our models and baselines are summarized in Table 1. Our CODEGEN-NL models (350M, 2.7B, 6.1B) outperform or perform on par with the respective GPT-NEO and GPT-J models. Further training CODEGEN-NL on multilingual programming language data (BIGQUERY) leads to CODEGEN-MULTI. The multilingual CODEGEN models outperform the models trained on THEPILE (GPT-NEO, GPT-J, CODEGEN-NL) by a large margin. We then finetune CODEGEN-MULTI on a Python-only dataset (BIGPYTHON), resulting in CODEGEN-MONO. The program synthesis capacity is improved substantially. Therefore, the Python program synthesis capacity enhances as the amount of Python training data increases. For almost all models, as expected, increasing the size of the model improves overall performance.

Our Python-monolingual CODEGEN models have competitive or improved performance, compared to the current state-of-the-art models, Codex. CODEGEN-MONO 2.7B underperforms CODEX 2.5B when $k = 100$ but outperforms it when $k \in \{1, 10\}$. While it is only half the size, our CODEGEN-MONO 6.1B demonstrates pass@k scores approaching those of the best-performing Codex, CODEX 12B. Our largest model CODEGEN-MONO 16.1B is competitive or outperforms it depending on $k$.

## 3.2 BETTER USER INTENT UNDERSTANDING YIELDS BETTER SYNTHESIZED PROGRAMS

The success of a program synthesis system highly depends on how well it understands user intent. When the system is based on a language model, the perplexity of problem prompts provides a proxy for the system's understanding of user intent specifications. A low perplexity of an intent specification under a model indicates that this intent specification is compatible with the knowledge learned by the model from the training data. We investigate whether better prompt understanding, with lower prompt perplexity as a proxy, leads to more functionally accurate programs.

We partition all problems into pass versus non-pass ones. A pass problem is one that at least one sample from 200 samples passes all test cases, while for a non-pass problem none of the 200 samples pass all test cases. We compute the average perplexity of the problem prompts of the pass problems and that of the non-pass ones, based on samples from CODEGEN-MONO models. The results are displayed in Table 2 (see Appendix F for the results on CODEGEN-NL and CODEGEN-MULTI). The prompts of the pass problems have lower perplexity than those of the non-pass ones. This finding implies that program synthesis is more likely to be successful when the user intent specification is understood better by the model. Indeed, some training data contains interleaved sequences of natural language comments and programs, where the comments describe the functionality of the following program. We thus speculate that user intent specifications similar to such a pattern would be better understood by the model, and hence lead to better program synthesis. Inspired by this pattern, we propose to specify user intent in multiple turns such that the model focus on a partial problem at a time, which would make user intent understanding by the model easier.

## 4 MULTI-TURN EVALUATION

In this section, we propose and study a multi-step program synthesis paradigm where program synthesis is decomposed into multiple steps and the system synthesizes a subprogram in each step. To examine such a paradigm, we first develop a Multi-Turn Programming Benchmark (MTPB). MTPB consists of 115 problems written by experts, each of which includes a *multi-step* descriptions in natural language (*prompt*). To solve a problem, a model needs to synthesize functionally correct subprograms (1) following the description at the current step and (2) considering descriptions and synthesized subprograms at previous steps (*e.g.*, correct backreference of functions and/or variables defined in the previous steps). An illustrative example is shown in Figure 1.

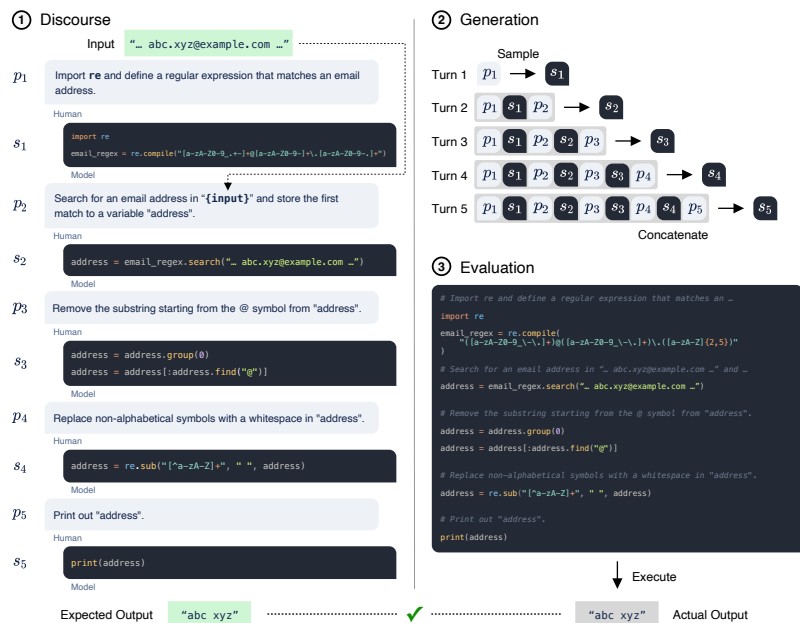

Figure 1: An illustrative example for the Multi-Turn Programming Benchmark, performing the task of extracting the user name of an email address. ① Each problem consists of prompts $p_i$ and unit tests, where some prompts include templates (*i.e.* {input}) that are filled with test case inputs before it is fed to the model. In the displayed example, the input is a string containing abc.xyz@example.com, which replaces {input} in $p_2$, and the expected output is abc xyz. ② Our model conditions on the concatenation of interleaved past prompts and *generated responses*. ③ Generated responses from each turn are concatenated and executed, where the output is compared to the answer.

## 4.1 Benchmark Construction

We (4 authors) start by defining[4] a set of 115 problems requiring a diverse range of programming knowledge, including math, array operations, string manipulations, algorithms, data science, and problems that require other knowledge, such that the number of problems in each category is roughly balanced.[5] For each problem, we construct a triplet consisting of multi-turn prompts $P$, test case inputs $I$, and test case outputs $O$. Multi-turn prompts $P$ are designed following the two constraints: (1) the problem is decomposed into 3 or more turns, (2) a single turn cannot be attributed to solving the problem. For example, implementing a linear regression model could be phrased as "Perform linear regression on x and y". Since the main task is fully expressed in this prompt, understanding this prompt is sufficient to perform the task. We avoid such cases via manual inspection and distribute problem-solving over turns. Together with the prompts, we task the problem author to prepare 5 sets of test case inputs $I$ and outputs $O$ to evaluate model outputs with functional correctness. To reduce wrongly rewarding false positive solutions that give meaningless programs but pass the tests, we examine and revise such cases to ensure the test quality.

Unlike HumanEval for which models are expected to complete a partially defined function, MTPB problems only provide the prompts, thereby models have to generate the solution from scratch.[6] While the free-form generation may allow for more potential solutions, the lack of an entry point to provide test case inputs makes it challenging to test the generated code on diverse test cases. To overcome this challenge, we instead embed test case inputs within prompts. Specifically, prompts are written with Python's formatted string[7] where input values are substituted for the variable name when a specific test case is applied to the problem. For example, a prompt, "Define a string named 's'

---

[4]Problem writing was performed in a closed book format, *i.e.* we are not allowed to consult with online resources while writing the problems.

[5]See Appendix D for a complete listing.

[6]To guide sampling in Python, we prefix the prompt with: # Import libraries.\n import numpy as np.

[7]https://docs.python.org/3/reference/lexical_analysis.html#f-strings

| Data | Model | Pass Rate$^\uparrow$ [%] | | | | |
|---|---|---|---|---|---|---|
| | | 350M | 2.7B | 6.1B | 16.1B | - |
| THEPILE | GPT-NEO & GPT-J | 0.79 | 8.17 | 18.86 | - | - |
| THEPILE | CODEGEN-NL | 0.23 | 15.31 | 19.37 | 30.33 | - |
| BIGQUERY | CODEGEN-MULTI | 4.09 | 20.82 | 25.51 | 26.27 | - |
| BIGPYTHON | CODEGEN-MONO | 16.98 | 38.72 | 43.52 | 47.34 | - |
| - | code-cushman-001 | - | - | - | - | 56.77 |
| - | code-davinci-001 | - | - | - | - | 55.28 |
| - | code-davinci-002 | - | - | - | - | 59.86 |

Table 3: Evaluation results on the Multi-Turn Programming Benchmark. The multi-turn program synthesis performance varies as a function of model size (columns) and code data size (rows).

| Prompt | PPL$^\downarrow$ | | | | Pass Rate$^\uparrow$ [%] | | | |
|---|---|---|---|---|---|---|---|---|
| | 350M | 2.7B | 6.1B | 16.1B | 350M | 2.7B | 6.1B | 16.1B |
| Single-Turn | $13.92 \pm 1.89$ | $11.67 \pm 1.46$ | $10.58 \pm 1.20$ | $10.25 \pm 0.99$ | 5.75 | 25.43 | 28.48 | 38.74 |
| Multi-Turn | $10.09 \pm 0.62$ | $8.90 \pm 0.52$ | $8.18 \pm 0.43$ | $8.05 \pm 0.43$ | 16.98 | 38.72 | 43.52 | 47.34 |

Table 4: Comparison between multi- and concatenated single-turn specifications on perplexity (PPL) and program synthesis performance (as measured by pass rate) under CODEGEN-MONO models.

with the value {var}.", together with a test case input `var = 'Hello'` will be formatted into "Define a string named 's' with the value 'Hello'." Also see ① in Figure 1 for an example.

## 4.2 EXECUTION ENVIRONMENT AND SOLUTION EVALUATION

For execution, the history of pairs of prompts and generated completions is concatenated into a self-contained program (see ③ in Figure 1 for an example). The program is then executed in an isolated Python environment following the single-turn HumanEval benchmark (Chen et al., 2021). However, the problems in HumanEval are constructed in such a way that a known function signature is completed, thus invocation of the generated code under a set of functional unit tests is trivial. In our multi-turn case, no such entry point (or return value) is guaranteed to be generated. To circumvent the issue of a missing return signature (or value), the last prompt of the multi-turn problems in MTPB is always specified to print out the resulting state to the terminal. Then, the benchmark execution environment overloads the Python `print(args)` function and stores `args` on a stack. If the sampled code for the last prompt of a problem does not include the `print()` statement, which is a valid convention to print on the terminal in Python or specifically Jupyter notebooks, then the AST of the generated code will be mutated to inject an invocation of `print()`. Finally, a type-relaxed equivalence check (*e.g.*, an implicit conversion between lists and tuples) of `args` against the predefined gold output of the problem is performed to determine test failure or success.

## 4.3 MULTI-STEP PROGRAMMING CAPACITY SCALES WITH MODEL SIZE AND DATA SIZE

In this analysis, we investigate how the model size and data size affect the program synthesis capacity in a multi-turn paradigm. In the MTPB, each problem has 5 test cases and we sample 40 samples for each test case with each model, based on which the pass rate is computed for each problem. The MTPB evaluation results (average pass rate) for our CODEGEN models, baselines, and OpenAI Codex models[8] are shown in Table 3. Clearly, the performance on the MTPB improves as a function of the model size and data size. This suggests that the capacity of multi-step program synthesis scales as a function of the model size and data size. The models are simply trained with an autoregressive language modeling objective. While the model and the data scale up, multi-turn program synthesis capacity emerges, that is, the capacity to synthesize programs in a multi-turn fashion.

---

[8]Accessed on November 10th, 2022.

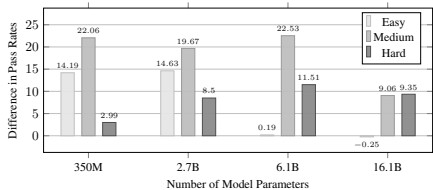

Figure 2: Difference in average pass-rate of problems in single-turn and multi-turn formulation over levels of problem difficulty. The improvement is sizable for most model sizes and difficulty levels, except for easy problems with larger models.

## 4.4 Better User Specification Understanding with Multi-turn Factorization

We hypothesize that multi-turn factorization enhances the model's understanding of user intent specifications, which in turn lead to higher program synthesis capacity. To test this hypothesis, we form a single-turn counterpart of multi-turn specifications by concatenating each specification into a single turn. As discussed in Section 3.2, we adopt the prompt perplexity as a proxy for user intent understanding. Thus, we compare the perplexity of the multi-turn prompts and that of the concatenated single-turn prompts under the four CODEGEN-MONO models.

The average perplexity (see Appendix E for the calculation details) over all the problems in the MTPB is displayed in the left panel of Table 4. For all models, the single-turn specification has a higher average perplexity than the multi-turn specification. It implies that the multi-turn user specifications can be better understood by the models. We notice that the average perplexity for both multi-turn and single-turn intent specifications under larger models is slightly lower than that under smaller models, indicating that the larger ones understand the user intent better than the smaller ones.

We compare the program synthesis pass rate with the multi-turn prompts to that with the concatenated single-turn prompts. The results are shown in the right panel of Table 4. Multi-turn specifications lead to close to or more than 10 percentage points over single-turn specifications for all model sizes. Together with the perplexity analysis above, it appears that factorizing a user specification into multiple steps and leveraging the emerged capacity of large language models allow them to digest the specification more easily and synthesize programs more successfully.

Furthermore, we categorize the problems by difficulty level based on their average pass rates ("hard" with less than 30%, "easy" with larger than 70%), and examine the interaction effect between difficulty level and model size on the improvement by multi-turn factorization. See the results in Figure 2. Across almost all model sizes and difficulty levels, multi-turn prompts lead to significant improvement over single-turn prompts and most improvements are nearly or higher than 10 percentage points. Interestingly, the larger models (6.1B and 16.1B) are invariant to multi-turn factorization for easy problems (see the two short bars, $0.19\%$ and $-0.25\%$, in Figure 2). This implies that when the problems can be easily understood by the model (due to the combined effect of easiness of the problems and the high capacity of larger models), it is not necessary or beneficial to factorize the specifications. This is in fact consistent with our motivating assumption that factorizing complicated specifications would ease problem understanding and improve program synthesis.

## 4.5 Qualitative Examples

To further understand the differences in model behavior over model sizes, we examine cases where large models have contrasting performances to smaller models. We specifically select problems for which CODEGEN-MONO 16.1B and CODEGEN-MONO 2.7B show a significant discrepancy in performance. On problems where CODEGEN-MONO 16.1B performed significantly worse compared to CODEGEN-MONO 2.7B, we observe that the larger model becomes inflexible due to taking the prompt literally. For example, initializing a number always results in an integer, despite the prompt asking to cast into a string (Figure 3), or the "return" keyword in a prompt triggers a function definition while the intent is to directly generate an executable program (Figure 4). However in general, larger-scale models overcome mistakes due to prompt misinterpretation by smaller models, including assigning multiple variables at the same time (Figure 5) or understanding the concept of any comparison (Figure 6).

## 5 RELATED WORK

**Program Synthesis**    While program synthesis has a long history, two inherent challenges remain unsolved: (1) intractability of the program space and (2) difficulty in accurately expressing user intent (Manna & Waldinger, 1971; Gulwani et al., 2017). A large body of prior research attempted to address (1) by exploring methods like stochastic search techniques (Parisotto et al., 2017; Schkufza et al., 2013) and deductive top-down search (Gulwani, 2011; Polozov & Gulwani, 2015). However, the scalability of these approaches is still limited. User intent can be expressed with various methods: formal logical specifications, input-output examples, and natural language descriptions. Complete and formal specifications require too much effort, while informal ones like input-output examples often under-specify problems (Gulwani, 2011). Well-learned conditional distribution and language understanding capacity owing to the large-scale model and data allows for efficient solutions for these two challenges. Several works investigate converting conversational intents into programmable representations, such as SQL (Yu et al., 2019a;b) or dataflow graph (Andreas et al., 2020). Our proposed benchmark requires the generation of Python, which is more general and complex.

**Large Language Models**    Transformers capture dependency among sequence elements through attention mechanism (Bahdanau et al., 2014) and are highly scalable. It has been successfully applied to natural language processing (Devlin et al., 2019; Lewis et al., 2020; Raffel et al., 2020), computer vision (Dosovitskiy et al., 2021), and many other areas (Oord et al., 2018; Jumper et al., 2021). Prior works, such as CuBERT (Kanade et al., 2020), CodeBERT (Feng et al., 2020), PyMT5 (Clement et al., 2020), and CodeT5 (Wang et al., 2021), have applied transformers towards code understanding but these mostly focus on code retrieval, classification, and program repair. Several recent and concurrent efforts explore using large language models for program synthesis (Chen et al., 2021; Austin et al., 2021; Li et al., 2022; Fried et al., 2022) and its effectiveness (Vaithilingam et al., 2022). While they focus on generating code in a single turn, we propose to factorize the specifications into multiple turns and demonstrate that it is highly effective to improve synthesis quality. It is worth pointing out that Austin et al. (2021) explored refining the code in multiple iterations, but it is essentially a single-turn approach since a complete program is produced in every single turn. Prompting pre-trained language models with intermediate information to improve task performance has attracted interest (Nye et al., 2021; Wei et al., 2022). Our proposed MTPB also allows the model to leverage past turns as context.

**Benchmarks for Program Synthesis**    To quantitatively evaluate program synthesis models, several benchmarks have been proposed with different input forms. A popular input forms include preceding code in the same line (Raychev et al., 2016), pseudo-code (Kulal et al., 2019), a docstring and function signature (Chen et al., 2021), or problem description (Hendrycks et al., 2021). In most of those cases, only directly relevant input information is given to the model. In contrast, a few previous works instantiate benchmarks that measure the ability to generate programs given surrounding program context beyond the target program, such as variables and other methods (Iyer et al., 2018) or alternating "cells" of preceding code and text blocks (Agashe et al., 2019), while the primary focus is to generate the target program itself. We propose a new benchmark that requires a progressive generation of subprograms through multi-turn prompts.

## 6 CONCLUSION

We study program synthesis with large causal language models trained on large corpora of code data. The capacity to understand long context and generate coherent responses emerges from the simple language modeling as the model size and data size scale up. Leveraging this capacity and observing that better user intent understanding leads to better program synthesis, we propose a multi-step program synthesis approach in which program synthesis is achieved through a multi-turn specification and code generation. Moreover, we develop the Multi-Turn Programming Benchmark (MTPB) to investigate our models' capacity on synthesizing programs in such a multi-step paradigm. Our experiments show that the multi-step program synthesis capacity scales as a function of the model size and data size. The intent specifications, which are specified in multiple steps, are digested more easily by the models and lead to more accurate program synthesis. We open-source the training code and the model checkpoints to facilitate future research and practical applications in this area.

## BROADER IMPACT AND ETHICAL CONSIDERATIONS

All variants of CODEGEN are firstly pre-trained on the Pile, which includes a small portion of profane language. Focusing on the GitHub data that best aligns our expected use case of program synthesis, Gao et al. (2020) report that 0.1% of the data contained profane language, and has sentiment biases against gender and certain religious groups. Thus, while we did not observe in our samples, CODEGEN may generate such content as well. In addition to risks on natural language outputs (*e.g.*, docstrings), generated programs may include vulnerabilities and safety concerns, which are not remedied in this work. Models should not be used in applications until being treated for these risks.

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

## A MODEL TRAINING

To evaluate the emergence of multi-turn program synthesis capabilities under scaling laws, we adopt standard transformer-based autoregressive language models, varying (1) the number of model parameters (350M, 2.7B, 6.1B, 16.1B) and (2) the number of tokens of programming languages in the training corpora. For scaling the models, a custom library JAXFORMER for training large language models on TPU-v4 hardware was developed and will be released as open source, including the trained model weights.

### A.1 DATASETS

| Dataset | Language | Raw Size | Final Size | Final Tokens |
|---|---|---|---|---|
| THEPILE | Natural Language | 825.18 GiB | 1159.04 GiB | 354.7B |
| | Code | 95.16 GiB | 95.16 GiB | 31.6B |
| BIGQUERY | C | 1772.1 GiB | 48.9 GiB | 19.7B |
| | C++ | 205.5 GiB | 69.9 GiB | 25.5B |
| | Go | 256.4 GiB | 21.4 GiB | 9.6B |
| | Java | 335.1 GiB | 120.3 GiB | 35.4B |
| | JavaScript | 1282.3 GiB | 24.7 GiB | 9.7B |
| | Python | 196.8 GiB | 55.9 GiB | 19.3B |
| BIGPYTHON | Python | 5558.1 GiB | 217.3 GiB | 71.7B |

Table 5: Approximate statistics for training corpora along the pre-processing steps.

For each dataset, the pre-processing shares the following steps: (1) filtering, (2) deduplication, (3) tokenization, (4) shuffling, and (5) concatenation. For details on THEPILE, we refer to Gao et al. (2020). For BIGQUERY and BIGPYTHON, in (1) files are filtered by file extension, and files with average lines length of <100 characters, a maximum line length of $1,000$, and >90% of the characters being decimal or hexadecimal digits are removed. For (2), exact duplicates based on their SHA-256 hash are removed, which amounts to a substantial portion of the raw data due to forks and copies of repositories. For (3), the BPE vocabulary of GPT-2 is extended by special tokens representing repeating tokens of tabs and white spaces. In the multi-lingual setting of BIGQUERY, a prefix is prepended to indicate the name of the programming language. For (4), each year of data is randomly shuffled. For (5), sequences are concatenated to fill the context length of $2,048$ tokens with a special token as a separator. Table 5 summarizes the statistics of the training corpora.

CODEGEN-NL models are randomly initialized and trained on THEPILE. CODEGEN-MULTI models are initialized from CODEGEN-NL and then trained on the BIGQUERY. CODEGEN-MONO models are initialized from CODEGEN-MULTI and then trained on BIGPYTHON.

### A.2 MODELS

Our models are autoregressive transformers with the regular next-token prediction language modeling as the learning objective. The family of CODEGEN models is trained in various sizes with 350M, 2.7B, 6.1B, and 16.1B parameters. The first three configurations allow for direct comparison with open-sourced large language models trained on text corpus, GPT-NEO (350M, 2.7B) (Black et al., 2021) and GPT-J (6B) (Wang & Komatsuzaki, 2021). See Table 6 in Appendix A for model specifications.

The architecture follows a standard transformer decoder with left-to-right causal masking. For the positional encoding, we adopt rotary position embedding (Su et al., 2021). For the forward pass, we execute the self-attention and feed-forward circuits in parallel for improved communication overhead following Wang & Komatsuzaki (2021), that is, $x_{t+1} = x_t + \text{mlp}(\ln(x_t + \text{attn}(\ln(x_t))))$ is altered to $x_{t+1} = x_t + \text{attn}(\ln(x_t)) + \text{mlp}(\ln(x_t))$ for which the computation of self-attention, $\text{attn}()$, and feed-forward, $\text{mlp}()$, with layer-norm, $\ln()$, is simultaneous. The architecture and hyper-parameter choices were optimized specifically for the hardware layout of TPU-v4.

| Model | Dataset | Hyper-parameter | 350M | 2.7B | 6.1B | 16.1B |
|---|---|---|---|---|---|---|
| CODEGEN | | Number of layers | 20 | 32 | 33 | 34 |
| | | Number of heads | 16 | 32 | 16 | 24 |
| | | Dimensions per head | 64 | 80 | 256 | 256 |
| | | Context length | 2,048 | 2,048 | 2,048 | 2,048 |
| | | Batch size | 500k | 1M | 2M | 2M |
| | | Weight decay | 0.1 | 0.1 | 0.1 | 0.1 |
| CODEGEN-NL | THEPILE | Learning rate | 3.0e−4 | 1.6e−4 | 1.2e−4 | 0.9e−4 |
| | | Warm-up steps | 3k | 3k | 3k | 3k |
| | | Warm-up / Total steps | 350k | 350k | 350k | 350k |
| CODEGEN-MULTI | BIGQUERY | Learning rate | 1.8e−4 | 0.8e−4 | 0.4e−4 | 0.5e−4 |
| | | Warm-up steps | 3k | 3k | 3k | 3k |
| | | Total steps | 150k | 150k | 150k | 150k |
| CODEGEN-MONO | BIGPYTHON | Learning rate | 1.8e−4 | 0.8e−4 | 0.4e−4 | 0.5e−4 |
| | | Warm-up steps | 3k | 3k | 3k | 3k |
| | | Total steps | 150k | 150k | 150k | 150k |

Table 6: Hyper-parameters for model specification and optimization for the family of CODEGEN models.

## A.3 TRAINING

The scaling of large language models requires data and model parallelism. Google's TPU-v4 hardware with a high-speed toroidal mesh interconnect naturally allows for efficient parallelism. To efficiently utilize the hardware, the training of the models is implemented in JAX (Bradbury et al., 2018). For parallel evaluation in JAX the $pjit()$[9] operator is adopted. The operator enables a paradigm named single-program, multiple-data (SPMD) code, which refers to a parallelism technique where the same computation is run on different input data in parallel on different devices.[10] Specifically, $pjit()$ is the API exposed for the XLA SPMD partitioner in JAX, which allows a given function to be evaluated in parallel with equivalent semantics over a logical mesh of compute.

Our library JAXFORMER recruits a designated coordinator node to orchestrate the cluster of TPU-VMs[11] with a custom TCP/IP protocol. For data parallelism, the coordinator partitions a batch and distributes the partitions to the individual TPU-VMs. For model parallelism, two schemes for the sharding of model parameters are supported: (1) Intra-TPU-VM, where parameters are sharded across MXU cores[12] inside a physical TPU-v4 board and replicated across boards following Shoeybi et al. (2019); Wang & Komatsuzaki (2021); (2) Inter-TPU-VM, where parameters are sharded across TPU-v4 boards and activations are replicated following Rajbhandari et al. (2020).

Both intra-TPU-VM and inter-TPU-VM sharding schemes are implemented based on our specific `pjit()` a logical mesh specification $(r, p, c)$ with $r$ replicas of the parameters, $p$ partitions of the parameters, and $c$ logical cores per board over $n_b$ TPU boards with each $n_c$ logical cores such that $d \times p = n_b$ and $r \times p \times c = n_b \times n_c$.

The intra-TPU-VM scheme is adopted for models of size of less or equal to 6B parameters, the total amount of model and optimizer parameters fit into the combined HBM memory of a single TPU-v4 board. For instance, a TPU-v4-512 slice with $n_b = 64$ and $n_c = 4$ would be configured as $(r, p, c) = (64, 1, 4)$. That is, the parameters are being replicated across $r = 64$ boards with $p = 1$ total inter-board partitions and intra-board parallelism across $c = 4$ logical chips. In this configuration, the mean gradient is accumulated across boards via `with_sharding_constraint()`, effectively emulating the behavior of the `xmap()`[13] operator.

---

[9] https://jax.readthedocs.io/en/latest/_modules/jax/experimental/pjit.html

[10] https://jax.readthedocs.io/en/latest/jax-101/06-parallelism.html

[11] https://cloud.google.com/blog/products/compute/introducing-cloud-tpu-vms

[12] Specifically, 4 TPU-v4 chips (*i.e.*, 8 physical which amount 4 logical or virtual MXU cores).

[13] https://jax.readthedocs.io/en/latest/_autosummary/jax.experimental.maps.xmap.html

The inter-TPU-VM scheme is adopted for models exceeding the size of 6B parameters for which the model and optimizer parameters have to be sharded across TPU-v4 boards. For instance, a TPU-v4-512 slice with $n_b = 64$ and $n_c = 4$ would be configured as $(r, p, c) = (1, 64, 4)$. For larger slices such as TPU-v4-1024 with $n_b = 128$, one may introduce redundancy in the parameter sharding, *e.g.*, $(r, p, c) = (2, 64, 4)$. In this configuration, the activations are replicated across boards via with_sharding_constraint(). Moreover, $(r, p, c)$ allows for backwards compatibility for the logical hardware layout transition from TPU-v3 with $c = 8$ to TPU-v4 with $c = 4$ by adjusting $p$ without the need for re-sharding.

For the optimization, Table 6 summarizes the hyper-parameters. We adopt the Adam (Kingma & Ba, 2015) optimizer with $(\beta_1, \beta_2, \epsilon) = (0.9, 0.999, 1e{-}08)$ and global gradient norm clipping (Pascanu et al., 2013) of 1.0. The learning rate function over time follows GPT-3 (Brown et al., 2020) with warm-up steps and cosine annealing. In summary, we mainly adopted the GPT-3 reference configurations with minor variations accounting for TPU optimizations. We did not have the compute capacity to optimize these hyper-parameters further.

# B  PASS@$k$ ESTIMATOR

We use the unbiased estimator proposed in Chen et al. (2021) to compute pass@$k$. For each task, $n \geq k$ samples are sampled. In particular, we use $n = 200$ and $k \leq 100$. Suppose $c$ is the number of correct samples, among the $n$ samples, which pass all the unit tests. Then the unbiased estimator is defined as follows:

$$\text{pass@}k = \mathbb{E}_{\text{Problems}} \left[ 1 - \frac{\binom{n-c}{k}}{\binom{n}{k}} \right] \tag{1}$$

Directly computing this estimator is numerically unstable. We use the numerically stable numpy implementation introduced by Chen et al. (2021).

# C  TYPE-RELAXED EQUIVALENCE CHECK FOR MTPB EVALUATION

We perform the following type-relaxation before assessing the equivalence between model outputs and the expected outputs.

- Convert numpy arrays into correspondingly typed lists of standard types (*e.g.* np.int32 will be cast to int).
- pandas series are converted and compared in numpy array format.
- For the rest, model outputs are cast into the type of gold standard outputs.
- Floating numbers are compared with $\varepsilon = 1e^{-6}$ as the tolerance threshold.

# D LIST OF MTPB PROBLEMS

| Problem Name | Problem Description | Category |
|---|---|---|
| Sandwich string | Append a string in the middle of another string. | string |
| Normalize integer list | Normalize a list of positive integers and print formatted percentages. | math |
| Convert time | Convert units of time. | math |
| Squared Fibonacci | Print the squared Fibonacci numbers. | math |
| Compare counts | Compare the count of positive and negative numbers in a given list. | array |
| Pandas mean | Construct and compute the mean of a pandas DataFrame. | D.S. |
| Fizz buzz | Solve the fizz buzz problem. | Algo. |
| Bi-grams | Print the bi-grams of a sentence. | string |
| Top note | Print the name with top note out of a dictionary. | dict |
| Hex to binary | Convert hex to binary and reverse. | math |
| Invert dict | Detect an inversion of a given dictionary. | dict |
| Class definition | Create a POJO class. | class |
| Longest number | Print the longest number. | math |
| Linear regression | Fit linear regression model with specified function and sk-learn. | D.S. |
| Encrypt and decrypt | Rotate alphabet for encryption, then reverse the operation. | Algo. |
| Dedup custom objects | Implement a class with __hash__ and obtain a count unique objects. | class |
| Drunken python | Convert between integer and string without using built-in functions. | string |
| Morse code | Encode a string into morse code given its conversion rule. | Algo. |
| Two-sum | Implement the two-sum problem on a given input pair. | Algo. |
| k-means | Implement and run k-means on sampled points. | D.S. |
| Even odd sum | Print the sum of even and odd numbers in a list. | math |
| Shift zeros | Move all the zeros in a list to the right. | array |
| Bootstrap 95% CI | Calculate the bootstrap 95% confidence interval of an array. | D.S. |
| Sum even digits | Sum the even digits between two numbers. | math |
| Min-max diff | Compute the difference between max and min numbers in a list. | array |
| Distinct chars | Print the sorted, case-insensitive unique characters of a string. | string |
| Longer string | Compare and print the longer string given two strings. | string |
| Sum float digits | Sum numbers before and after the decimal point of a float. | math |
| Count vowels | Count the number of vowels in a string. | string |
| Factorial | Compute the factorial of n. | math |
| Max edge triangle | Finds the maximum range of a triangle's third edge. | math |
| Factorial & remainder | Compute the factorial and its remainder when divided. | math |
| Sum polygon angles | Sum the angles in a polygon. | math |
| Sum string numbers | Add together two numbers represented in string. | string |
| Min-max sum | Sum the range from the minimum to the maximum of a list. | array |
| Vowel overlap | Find the number of overlapped vowels of two words. | string |
| Sum negative | Calculate the sum of negative numbers in a list. | math |
| Load dataset | Load from a file and print statistics. | D.S. |
| Char length list | Return a list of non-punctuation character lengths from words. | string |
| Hex to RGB | Convert a six hexadecimal digit string into list of RGB values. | math |
| Majority vote | Check if a certain element is the majority of a given list. | array |
| Week later | Print the formatted date of a week later given a date. | string |
| Sorted word weights | Check if the list of word weights (sum of ASCII values) are sorted. | math |
| Create Palindrome | Sum pairs of adjacent digits until the number is palindrome. | string |
| Simulate Backspace | Apply the backspace characters in a string and print the modified. | string |
| Data manipulation | Manipulate a pandas DataFrame and split into train and test set. | D.S. |
| Sum non-overlap | Sum the integers in a (min, max) range that don't appear in a list. | array |
| Detect digits | Find if a string contains digits. | array |
| Cascading functions | Sequentially invoke function objects in a given list. | math |
| Pluralize duplicates | Pluralize duplicated words in a list. | dict |
| Highest altitude | Given relative altitudes , find the highest altitude | array |
| Truncate words | Truncate a sentence so that it contains k words | array |
| Single element | Find the elements that appear one time in an array | array |
| Remove elements | Remove all the occurrences of an element in an array | array |
| Check array sum | Check whether the sum of an array is equal to a given value | array |

Table 7: Problems in MTPB, showing the problem 1 to 55. D.S. and Algo. refers to data science and algorithm.

| Problem Name | Problem Description | Category |
|---|---|---|
| Merge sorted lists | Merge two sorted lists into one | Algo. |
| Maximum subarray | Find the max contiguous subarray and return the sum | Algo. |
| Max square root integer | Find the largest integer but smaller than the square root | Algo. |
| Longest word | Find the longest word in a word list | Algo. |
| Sum unique elements | Sum all the unique numbers in a list | Algo. |
| Diagonal sum | Compute the diagonal sum of a matrix | D.S. |
| Matrix condition number | Check condition number of a matrix is less than a threshold | D.S. |
| Matrix multiplication sum | Compute matrix multiplication sum of two matrices | D.S. |
| Matrix determinant | Compare two matrix determinants | D.S. |
| Log-sum-exp | Compute the log of sum exponential input | D.S. |
| K nearest points | Find the k nearest points to the origin | array |
| Longest common prefix | Find the longest common prefix of two strings | Algo. |
| Duplicate elements | Find duplicates in a list | array |
| First unique character | Find the first non-repeating character in a string | Algo. |
| Uncommon words | Find uncommon words in two sentences | Algo. |
| Average words length | Compute the average word length of a sentence | Algo. |
| Compare char freq | Compare the character frequencies in two strings | string |
| Reverse string | Reverse a string | string |
| Square Sum diff | Difference between the square of sum and the sum of squares | math |
| Cosine sim | Compute the cosine similarity between two vectors | math |
| Vector distance | Compare vector distances to the origin | math |
| Smallest standard dev. | Find the smaller standard deviation given two lists | D.S. |
| Smallest means | Find the smaller mean given two lists | D.S. |
| Coefficient of variation | Compute coefficient of variation given a list | D.S. |
| L1 norm | Compute the L1 norm given a list | D.S. |
| Z-statistic | Compute z-statistic given a list | D.S. |
| Move negatives | Move all negative elements in a list to the end | array |
| Remove alphabets | Remove alphabetical characters in a string | string |
| Largest norm | Find the largest norm among n-dimensional points | D.S. |
| F1 score | Given two arrays (pred, gold), calculate the F1 score | D.S. |
| Add Space | Add spaces before capital letters | string |
| Remove outlier | Remove data points in the tail (2sigma) of normal distribution | D.S. |
| Convert to categorical | Convert values into categorical variables | D.S. |
| Group by key | Group items in an array using a provided function | array |
| Max stock profit | Given an array of "prices", find the max profit | array |
| Sum positions | Sum of all position indices where a value appear | array |
| Find missing num | Find a missing number given a list and a max number | array |
| Common num in matrix | Common numbers among rows in a matrix | array |
| Sum Collatz | Obtain the sum of Collatz sequence starting from given number | Algo. |
| Cup swap | Name the location of a "ball" after cup swapping | Algo. |
| Reverse digits | Reverse digits in a number with a stack | Algo. |
| Calculate arrows | Calculate arrowheads left and right | Algo. |
| Check interval num | Check if the interval (max-min) is included in a list | Algo. |
| Length encoding | Encode a string by converting repeated chars with counts | string |
| Convert email | Use regex to match email addresses and remove special chars | string |
| Second largest | Print out the second largest element in an array | array |
| Largest prefix sum | Return the largest prefix sum in an array | array |
| Closest element to zero | Find the element which is the closest to 0 and print the distance | array |
| Consecutive unique char | Find the max length contiguous subarray with unique characters | string |
| Highest frequency char | Obtain the frequency of the most frequent character | string |
| Longest palindrome | Find the length of longest palindrome substring | string |
| Count primes | Calculate prime numbers in a range | Algo. |
| Rotate array | Rotate an array to the right k steps | Algo. |
| Partition equal sets | Check if an array can be split into two sets with equal sums | Algo. |
| Square root integer | Compute the integer part of square root | math |
| Plus 1 | Return the digits after an integer is added by 1 | math |
| Check square sum | Check whether one integer is a sum of two square numbers | math |
| Compare standard dev. | Determine whether standard deviation is less than 1 | D.S. |
| Matrix size | Calculate the sum of row and column numbers | D.S. |
| Diff mean and median | Calculate the difference between mean and median for an array | D.S. |

Table 8: Problems in MTPB, showing the problem 56 to 115. D.S. and Algo. refers to data science and algorithm.

# E   PERPLEXITY COMPUTATION FOR SINGLE- AND MULTI-TURN PROMPTS

Suppose $\{p_i\}_{i=1}^n$ is the set of prompts for a given problem, and $\{s_i\}_{i=1}^n$ are the $n$ sub-programs synthesized by a model $P_\theta$. Suppose $c_{i-1} = [p_1; s_1; ...; p_{i-1}; s_{i-1}]$ where $[\cdot\,;\cdot]$ indicates concatenation, the conditional probability of $p_i$ is $\mathrm{Prob}_i = P_\theta(p_i|c_{i-1})$, and then the perplexity for the multi-turn prompts is computed as

$$\mathrm{PPL}_{\mathrm{Multi-turn}} = \exp\left(-\frac{1}{m}\sum_{i=1}^n \log \mathrm{Prob}_i\right), \tag{2}$$

where $m$ is the total number of tokens of all prompts $\{p_i\}_{i=1}^n$. Suppose $c = [p_1; s_1; ...; p_n, s_n]$, then its probability is $\mathrm{Prob} = P_\theta(c)$, and the the perplexity for the single-turn prompts is computed as

$$\mathrm{PPL}_{\mathrm{Single-turn}} = \exp\left(-\frac{1}{m}\log \mathrm{Prob}\right). \tag{3}$$

# F   PERPLEXITY COMPARISON FOR CODEGEN-NL AND CODEGEN-MULTI

| CODEGEN-NL | 350M | 2.7B | 6.1B |
|---|---|---|---|
| Pass | 4.53 | 3.25 | 2.78 |
| Non-Pass | 4.96 | 3.87 | 3.65 |

Table 9: Average prompt perplexity$^\downarrow$ of CODEGEN-NL models on pass and non-pass problems.

| CODEGEN-MULTI | 350M | 2.7B | 6.1B |
|---|---|---|---|
| Pass | 4.78 | 3.82 | 3.82 |
| Non-Pass | 5.64 | 4.85 | 4.80 |

Table 10: Average prompt perplexity$^\downarrow$ of CODEGEN-MULTI models on pass and non-pass problems.

# G   ADDITIONAL BENCHMARK RESULTS

| Model | pass@1 | pass@10 | pass@100 |
|---|---|---|---|
| CODEGEN-NL 350M | 0.96 | 6.37 | 19.91 |
| CODEGEN-NL 2.7B | 5.34 | 24.63 | 48.95 |
| CODEGEN-NL 6.1B | 8.15 | 31.21 | 55.27 |
| CODEGEN-NL 16.1B | 10.92 | 38.43 | 62.76 |
| CODEGEN-MULTI 350M | 7.46 | 24.18 | 46.37 |
| CODEGEN-MULTI 2.7B | 18.06 | 45.80 | 65.34 |
| CODEGEN-MULTI 6.1B | 18.35 | 47.27 | 67.92 |
| CODEGEN-MULTI 16.1B | 20.94 | 51.61 | 70.02 |
| CODEGEN-MONO 350M | 14.59 | 41.49 | 63.00 |
| CODEGEN-MONO 2.7B | 27.31 | 59.19 | 74.24 |
| CODEGEN-MONO 6.1B | 32.48 | 64.20 | 76.81 |
| CODEGEN-MONO 16.1B | 35.28 | 67.32 | 80.09 |
| INCODER 6B | 21.30 | 46.50 | 66.20 |
| code-cushman-001 | 45.90 | 66.90 | 79.90 |
| code-davinci-001 | 51.80 | 72.80 | 84.10 |
| code-davinci-002 | 58.10 | 76.70 | 84.50 |

Table 11: Pass rates on Mostly Basic Python Problems (MBPP).

We also evaluated our models on Mostly Basic Python Problems (MBPP) (Austin et al., 2021). The results are displayed in Table 11. Following Chen et al. (2022), we sampled programs from the

sanitized MBPP for all of our models, with $n = 100$ and temperature$= 0.8$. The last four rows are from the aforementioned paper. In general we observe the consistent trend of improving the performance over different versions (NL, Multi, Mono), with our largest CODEGEN-MONO 16.1B approaching the results from code-cushman-001. While we do not know whether any of OpenAI models is the "Codex 12B" reported in Chen et al. (2021), we believe our model achieves reasonable results on MBPP as well. We also note that our CODEGEN-MONO 6.1B significantly outperformed INCODER 6B.

## H  GENERATED SAMPLES

### H.1  CASES WHERE CODEGEN-MONO 16.1B UNDER-PERFORMS

```python
# Import libraries.

import numpy as np

# Initialize a variable "stack" with an
    empty list, and "num" with 100 as a
    string.

stack = []
num = "100"

# For each chracter in "num", append the
    character to "stack".

for i in range(len(num)):
    stack.append(num[i])

# Assign an empty string to a variable "
    result", and concatenate characters
    popped from the last element of "stack"
    to "result" until "stack" is empty.

result = ""
while len(stack)!= 0:
    result += stack.pop()

# Cast "result" as integer and print it out.

print(int(result))
```

```python
# Import libraries.

import numpy as np

# Initialize a variable "stack" with an
    empty list, and "num" with 100 as a
    string.

stack = []
num = 100

# For each chracter in "num", append the
    character to "stack".

for i in num:
    stack.append(i)

# Assign an empty string to a variable "
    result", and concatenate characters
    popped from the last element of "stack"
    to "result" until "stack" is empty.

result = ""
while len(stack) > 0:
    result += stack.pop()

# Cast "result" as integer and print it out.

print(int(result))
```

CODEGEN-MONO 2.7B                                CODEGEN-MONO 16.1B

Figure 3: Generated samples for "Reverse digits" problem. While being simplistic, we often observe that CODEGEN-MONO 16.1B assumes the type when assigning a number to a variable. Here on the right, despite being explicitly specified to initialize a number as a string, the larger model fails to do so. This causes an error in the following turn when the code attempts to iterate over num.

```
# Import libraries.

import numpy as np

# Assign the matrix "[[3, 2], [2, 3]]" to a
    variable named "my_matrix".

my_matrix = [[3, 2], [2, 3]]

# Assign the number "1" to a variable named
    "t".

t = 1

# Compute the condition number of my_matrix
    and store as result.

result = np.linalg.cond(my_matrix)

# Check whether the result is smaller than t
    . If yes, return "True", otherwise
    return "False".

if result < t:
    print("True")
else:
    print("False")
```

```
# Import libraries.

import numpy as np

# Assign the matrix "[[3, 2], [2, 3]]" to a
    variable named "my_matrix".

my_matrix = [[3, 2], [2, 3]]

# Assign the number "1" to a variable named
    "t".

t = 1

# Compute the condition number of my_matrix
    and store as result.

condition_number = np.linalg.cond(my_matrix)

# Check whether the result is smaller than t
    . If yes, return "True", otherwise
    return "False".

def check_condition_number(condition_number,
    t):
    if condition_number < t:
        return True
    else:
        return False
```

CODEGEN-MONO 2.7B                    CODEGEN-MONO 16.1B

Figure 4: Generated samples for "Matrix condition number" problem. Both models generate programs in the same scope until the last turn, where CODEGEN-MONO 16.1B generates in the form of function definition. Noticeably, the larger model generates `return` keyword that appears in the prompt, but it is only valid within a function.

## H.2 CASES WHERE CODEGEN-MONO 16.1B OUTPERFORMS

```python
# Import libraries.

import numpy as np

# Define a variable "pos" with "B", "swap"
    with ['AC', 'CA'].

pos = ['B','swap']

# Write a function "move" that takes two
    strings x and y as input, and replace
    any appearance of x in y with an empty
    string, then return y.

def move(x,y):
    return y.replace(x,'')

# For each element in "swap", if it contains
    "pos", call "move" on "pos" and the
    current element and store the result to
    "pos".

def swap(swap):
    for i in range(len(swap)):
        if swap[i] == 'pos':
            swap[i] = move(pos[0],pos[1])
    return swap

# Print out "pos".

print(pos)
```

CODEGEN-MONO 2.7B

```python
# Import libraries.

import numpy as np

# Define a variable "pos" with "B", "swap"
    with ['AC', 'CA'].

pos = "B"
swap = ['AC', 'CA']

# Write a function "move" that takes two
    strings x and y as input, and replace
    any appearance of x in y with an empty
    string, then return y.

def move(x, y):
    return y.replace(x, "")

# For each element in "swap", if it contains
    "pos", call "move" on "pos" and the
    current element and store the result to
    "pos".

for i in swap:
    if pos in i:
        pos = move(pos, i)

# Print out "pos".

print(pos)
```

CODEGEN-MONO 16.1B

Figure 5: Generated samples for "Cup swap" problem. Here, CODEGEN-MONO 2.7B mistakenly treats "swap" as a part of a value t o be assigned, while CODEGEN-MONO 16.1B parses the intent correctly and assigns two variables in parallel.

```python
# Import libraries.

import numpy as np

# Initialize the variable named lst1 with a
    list ['abc', 'abc10'].

lst1 = ['abc', 'abc10']

# Create a function called num_in_str() to
    check whether a string contains a number
    .

def num_in_str(string):
    for i in string:
        if i.isnumeric():
            return True
        else:
            return False

# Call the function num_in_str() to find
    strings in lst1 that have numbers and
    assign them to a list named lst2

lst2 = []

for i in lst1:
    if num_in_str(i):
        lst2.append(i)

# Print out lst2

print(lst2)
```

```python
# Import libraries.

import numpy as np

# Initialize the variable named lst1 with a
    list ['abc', 'abc10'].

lst1 = ['abc', 'abc10']

# Create a function called num_in_str() to
    check whether a string contains a number
    .

def num_in_str(s):
    return any(i.isdigit() for i in s)

# Call the function num_in_str() to find
    strings in lst1 that have numbers and
    assign them to a list named lst2

lst2 = [i for i in lst1 if num_in_str(i)]

# Print out lst2

print(lst2)
```

CODEGEN-MONO 2.7B                     CODEGEN-MONO 16.1B

Figure 6: Generated samples for "Detect digits" problem. CODEGEN-MONO 2.7B fails to implement the num_in_str, resulting in checking only the first character. In contrast, CODEGEN-MONO 16.1B successfully utilizes any function to scan all the characters in the given string.

## I   ADDITIONAL ANALYSES ON MTPB

We conducted additional analyses to illustrate the relationship generated program length and pass rate and showed the results in Figure 7, Figure 8, and Figure 9. The relationship between generated program length and prompt length is shown in Figure 10.

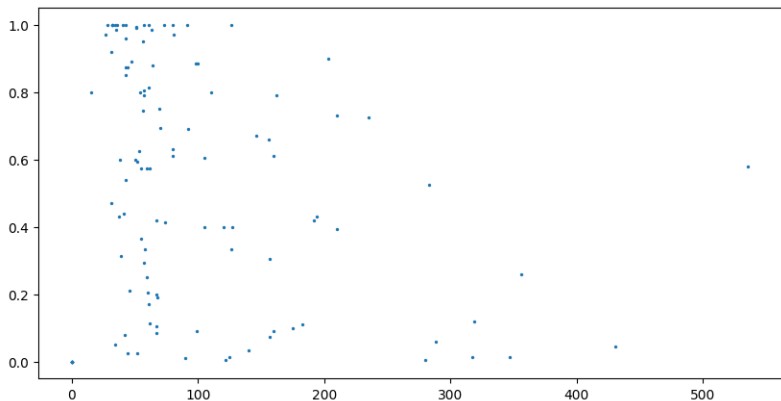

Figure 7: Maximum Length of Completion versus Pass Rate.

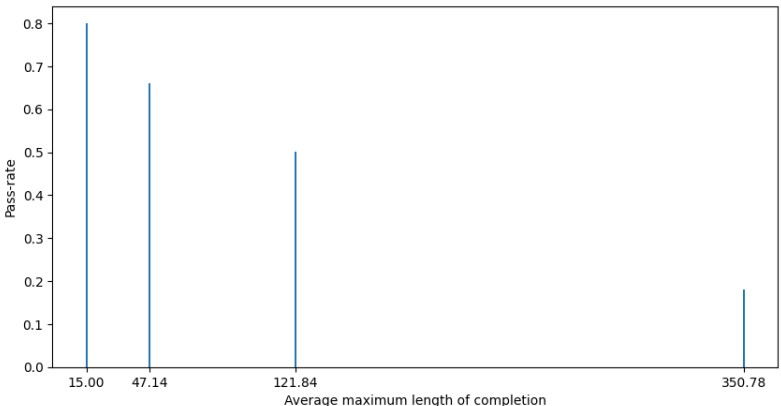

Figure 8: Maximum Length of Completion versus Pass Rate.

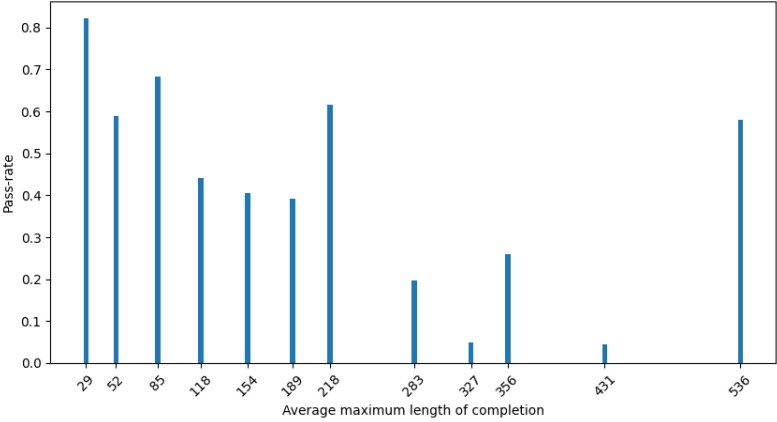

Figure 9: Maximum Length of Completion versus Pass Rate.

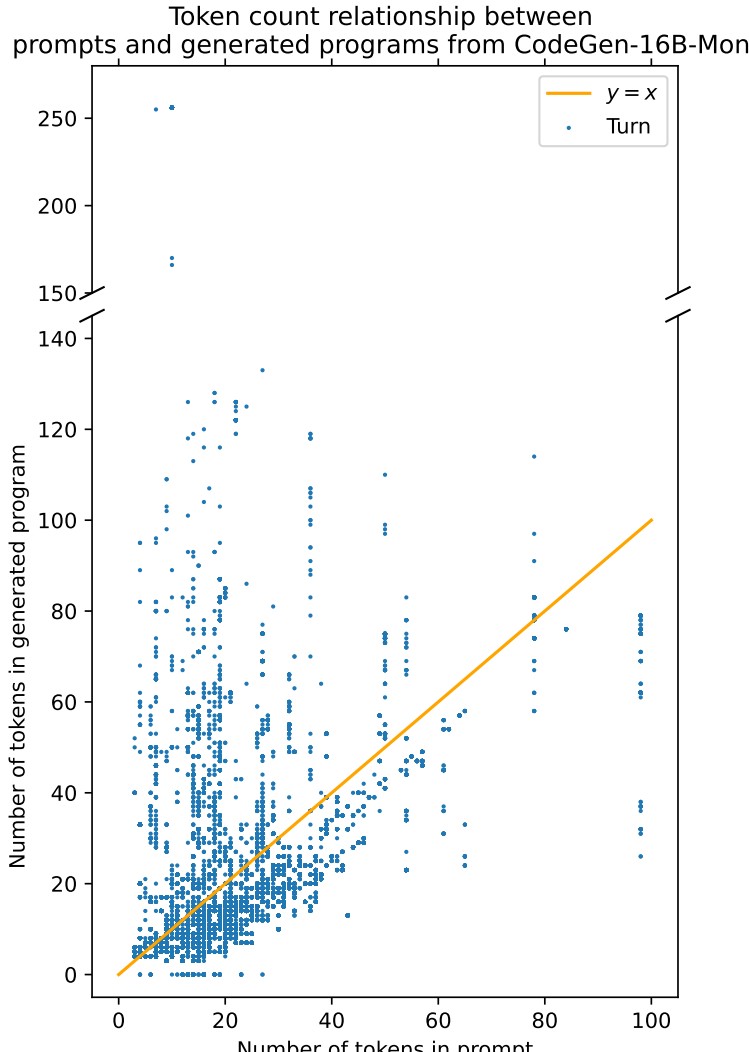

Figure 10: Prompt Length versus Generated Program Length.

