# OpenReview forum: "CodeGen: An Open Large Language Model for Code with Multi-Turn Program Synthesis"
_ICLR.cc/2023/Conference — ICLR 2023 notable top 25%_

### Official Review · Reviewer_6Afu · 2022-10-24

**Confidence:** 4
**Clarity, Quality, Novelty And Reproducibility:** See above
**Correctness:** 3
**Technical Novelty And Significance:** 3
**Empirical Novelty And Significance:** 3
**Recommendation:** 8

**Strength And Weaknesses:**

## Paper strengths and contributions
**Motivation and intuition**
The motivation for multi-turn code generation is convincing. The supervision of comments and programs is helpful for multi-turn program synthesis.

**Novelty**
The idea of utilizing weak supervision of interleaved patterns is intuitive and convincing. This paper presents an effective way to make use of this idea.

**Technical contribution**
Codegen for program synthesis seems effective, especially when the user wants to generate pieces of code from input/output examples or natural language descriptions.

**Clarity**
The overall writing is clear. The authors utilize figures and tables well to illustrate the ideas. Figure 1 clearly shows the code generation process.

**Related work**
The authors clearly describe the related prior works from both the perspectives of program synthesis, large language models, and benchmarks for program synthesis.

**Experimental results**
The presentation of the experimental results is clear. Mainly, Table 4 provides understandable results showing that multi-turn specifications achieve better performance compared to single-turn specifications.

**Reproducibility**
I believe reproducing the results is possible given the clear description provided in the main paper and the appendix.

## Paper weaknesses and questions

**Code comment analysis**
Some programmers like to write comments, while some are not. Also, different programmers write comments very differently. I suggest the authors investigate the effect of such differences. One very simple way could be to analyze the occurrence frequency of interleaved natural and programming patterns in the dataset. While it is hard to formally define meaningful comments, it would be insightful to at least calculate the document frequency of interleaved natural and programming language.

**Experiment setup**
In Table 3, the paper only compares the proposed method against GPT-Neo and GPT-J, which is not sufficient. The result will stand out to compare against Codex, the state-of-the-art program synthesis model.

**Summary Of The Paper:**

This paper trains large language models in sequence on three datasets, both natural language and programming language, and publishes the training library, JAXFORMER. The largest model outperforms the codex model on the HumanEval benchmark. The experiments show that the passing problem has lower prompt perplexity on the HumanEval benchmark. Some training data contains interleaved sequences of natural language comments and programs. For example, programmers often explain the functionality of programs with comments written with code. This work hypothesizes that the pattern provides weak supervision of multi-turn code generation. To further investigate this, the paper proposes the multi-turn code generation framework and constructs an open-source benchmark, Multi-Turn Programming Benchmark, to evaluate the proposed framework. For each turn in the generation, the model condition on the concatenation of interleaved past prompts and generated responses. Multi-turn generation specification invariantly outperforms single-turn specification. The evaluation result on the constructed dataset supports the proposed hypothesis. The performance difference in multi-turn and single-turn specification over levels of difficulty and models shows that the improvement is sizable except for easy problems with larger models.

**Summary Of The Review:**

I vote for accepting this paper because its idea is original and contributes to this area while some improvement can be made (see above).

---

> ### Author Response · Authors · 2022-11-12
> **Reply from Authors**
>
> ### Q1. Occurrence frequency of interleaved NL & PL patterns
>
> Indeed, it would be an interesting analysis to calculate the frequency of the interleaved NL and PL patterns, or even better, to investigate the relationship between the interleaving frequency and the MTBP performance. However, we don’t find a good way to track the interleaving frequency. As you mentioned, comments can appear either before or after code. If we take into account docstrings, some of them also contain instructions or implementation plans. Docstrings are in the middle of code. Therefore, it seems not easy to track the interleaving frequency.
>
>
> ### Q2. Evaluate Codex on MTPB
>
> Thank you for the suggestion! We refer Codex Davinci’s results on HumanEval to (Chen et al., 2022). We additionally run Codex models via OpenAI API and report results for MTPB in the table below. We have revised Tables 1 and 3 in the revision accordingly.
>
> | Model            | Pass rate |
> | ---------------- | --------- |
> | CodeGen-16B-mono | 47.31     |
> | code-cushman-001 | 56.77     |
> | code-davinci-001 | 55.28     |
> | code-davinci-002 | 59.86     |
>
>
> In particular, we would like to emphasize that the CodeGen performance is relatively “close” to code-cushman-001, which presumably has been continuously improved since the release of the Codex model initially reported in HumanEval.
>
> Noteworthy, while the performance of all three Codex variants on MTPB is relatively close, the distribution of pass rates across individual problems highly varies between the model variants. That is, some model variants can entirely solve a subset of problems, which fails entirely for other variants of Codex. We contrast a subset of MTPB problems in the table below.
>
> |         | MTPB Passes      |  | |
> | ------- | ---------------- | ---------------- | ---------------- |
> | Problem | code-cushman-001 | code-davinci-001 | code-davinci-002 |
> | 1       | 40 / 200         | 146 / 200 | 98 / 200 |
> | 10      | 199 / 200        | 16 / 200 | 32 / 200 |
> | 100     | 6 / 200          | 43 / 200 | 110 / 200 |
> | 16      | 193 / 200        | 143 / 200 | 2 / 200 |
> | 97      | 2 / 200          | 9 / 200 | 197 / 200 |
> | 99      | 188 / 200        | 198 / 200 | 0 / 200 |
>
> Chen et al. (2022): Chen, Bei, et al. "Codet: Code generation with generated tests." arXiv preprint arXiv:2207.10397 (2022).

---

> > ### Comment · Reviewer_6Afu · 2022-11-25
> > **Re: Reply from Authors**
> >
> > I appreciate the author's rebuttal, which addresses some of my questions. After carefully reading other reviewers' comments, I have decided to keep my original score.

---

### Official Review · Reviewer_JBoh · 2022-10-25

**Confidence:** 4
**Correctness:** 3
**Technical Novelty And Significance:** 2
**Empirical Novelty And Significance:** 2
**Recommendation:** 8

**Clarity, Quality, Novelty And Reproducibility:**

The paper is fairly well-written and reproducible with the models being publicly available. Novelty is limited since there are already several pre-training/dataset collection papers in this area.

For the MTPB dataset, I don't fully understand the part about embedding the test case inputs within the prompts. Why was this approach chosen over HumanEval's style of learning to complete given a function definition? How do you deal with multiple test cases? How does inference and evaluation work with multiple test cases?

It will be useful to list Codex Davinci's performance in Table 1 (for completeness, even though it is a much bigger model). Similarly, it would be interesting to look at Codex's performance on the MTPB dataset.


**Strength And Weaknesses:**

**Strengths:**
- CodeGen model performs comparatively to Codex (12B) model and is open-sourced (unlike Codex). So it should help facilitate future research in this area of AI for code.

- The results on how various pretraining datasets impacts the coding performance of the models is very interesting.

**Weaknesses:**
- CodeGen model (single-turn) is only evaluated on a single dataset with only 164 problems. Why is the model not evaluated on other publicly available coding datasets such as MBPP and APPS?

- While the multi-turn dataset helps the models generate more correct code, I am not sure about its practical use cases. Would a user be able to specify the sub-tasks in the required format (such as defining appropriate variables) as in Figure 1?

- How does multi-turn task break down work for programs with control flow structures (for e.g. write code for binary search)? From the examples, it appears that the MTPB dataset treats small loops and branches as a single sub-task, but how can a user specify sub-tasks within a loop/branch?

**Summary Of The Paper:**

This paper presents new family of open-source code large language models called CodeGen of size upto 16B parameters. This model outperforms Codex 12B model on the HumanEval dataset. The paper also presents a new multi-turn programming dataset where the specification for a single task is divided into multiple sub-tasks so that the model can solve each sub-task independently to solve the full task.


**Summary Of The Review:**

The contribution with regards to the CodeGen model is helpful for the community (since it is open-sourced and performs similar to Codex 12B model), but there is limited novelty there.

The MTPB dataset contribution needs more explanation regarding its practical use cases and its limited capability to handle more complex programs.

POST REBUTTAL: most of my questions are answered and I am increasing my score to a 8.

---

> ### Author Response · Authors · 2022-11-12
> **Reply from Authors (3/3)**
>
> ### Q3. How does multi-turn task break down work for programs with control flow structures
>
> Our multi-turn paradigm currently operates within the same global scope. Because of this, complex control flows must be specified in one turn and generated in one code block (e.g., define a whole if-block). We agree that this constrains the type of programs that can easily be generated and limits practicality. A potential remedy to this problem is to allow users or models to change scopes (inward or outward) over the turns. In the case of Python where indentation controls the scope, a model could learn to indent/dedent the right amount to stay within or move out of the current scope. We leave this for future work.
>
> ### Q4. Clarify embed test cases
>
> Thank you for your question. We embedded test cases into the specification in order to closely follow the multi-turn paradigm end-to-end, where relevant values to perform the task are also a part of the specification. To accommodate for multiple test cases, a set of multi-turn prompts for each test case is passed to the model and sampled. This procedure involves n times the inference (n = number of test cases) compared to the evaluation setting for HumanEval. That being said, we believe a proper evaluation of multi-turn code generation includes input specifications and therefore embedded test cases are necessary.
>
> ### Q5. Evaluate Codex on MTPB
>
> Thank you for the suggestion! We refer Codex Davinci’s results on HumanEval to (Chen et al., 2022). We additionally run Codex models via OpenAI API and report results for MTPB in the table below. We have revised Tables 1 and 3 in the revision accordingly.
>
> | Model            | Pass rate |
> | ---------------- | --------- |
> | CodeGen-16B-mono | 47.31     |
> | code-cushman-001 | 56.77     |
> | code-davinci-001 | 55.28     |
> | code-davinci-002 | 59.86     |
>
>
> In particular, we would like to emphasize that the CodeGen performance is relatively “close” to code-cushman-001, which presumably has been continuously improved since the release of the Codex model initially reported in HumanEval.
>
> Noteworthy, while the performance of all three Codex variants on MTPB is relatively close, the distribution of pass rates across individual problems highly varies between the model variants. That is, some model variants can entirely solve a subset of problems, which fails entirely for other variants of Codex. We contrast a subset of MTPB problems in the table below.
>
> |         | MTPB Passes      |  | |
> | ------- | ---------------- | ---------------- | ---------------- |
> | Problem | code-cushman-001 | code-davinci-001 | code-davinci-002 |
> | 1       | 40 / 200         | 146 / 200 | 98 / 200 |
> | 10      | 199 / 200        | 16 / 200 | 32 / 200 |
> | 100     | 6 / 200          | 43 / 200 | 110 / 200 |
> | 16      | 193 / 200        | 143 / 200 | 2 / 200 |
> | 97      | 2 / 200          | 9 / 200 | 197 / 200 |
> | 99      | 188 / 200        | 198 / 200 | 0 / 200 |
>
> Chen et al. (2022): Chen, Bei, et al. "Codet: Code generation with generated tests." arXiv preprint arXiv:2207.10397 (2022).

---

> > ### Comment · Reviewer_JBoh · 2022-11-18
> > **Reply to the authors**
> >
> > Thanks for adding the new experimental results. They certainly increase my confidence in the models, so I am increasing my score to a 8.
> >
> > I am still a bit confused about embedding test cases (Q4). How do you check if one of the 40*5 programs that are generated is correct? Do you run each of these programs on all 5 test cases before declaring it as correct?
> >
> > Also, if values in the inputs are important for the model to solve the task, you could also just pass it as examples (all together instead of one by one) after the problem description and before the partial function definition (similar to the human eval dataset). But, I am happy with the rest of the paper, so I would consider this as a minor point that the paper can further improve on.

---

> > > ### Author Response · Authors · 2022-11-19
> > > **Reply from Authors**
> > >
> > > We are glad to hear about your positive feedback. Thank you for proposing these additional experiments in the first place!
> > >
> > > Regarding embedding test cases, each problem has 5 test cases and we generate 40 samples per test case. This results in 5*40 samples for which we check whether each of the samples passes the corresponding test case. The number of passed samples among the total 200 samples is the pass rate for this problem.
> > >
> > > Once more, thank you.

---

> > > > ### Comment · Reviewer_JBoh · 2022-11-21
> > > > **Thanks for the clarification**
> > > >
> > > > Thanks for the clarification. This is a slightly non conventional metric, so please add details regarding this metric to the paper for better reproducibility.

---

> > > > > ### Author Response · Authors · 2022-11-29
> > > > > **Reply from Authors**
> > > > >
> > > > > Yes, we will clarify and revise the details in the next revision. Thank you

---

> ### Author Response · Authors · 2022-11-12
> **Reply from Authors (2/3)**
>
> ### Q2. Practical use cases of MTBP (see Q4 in Reviewer 2)
>
> While the example in Figure 1 involves verbose specifications which gives the impression that one has to specify in a right way, we would like to clarify that there is no fixed format required for certain operations. Many problems in our benchmark express the specification at high-level (e.g., please see k-means problem, which expects the model to generate a valid implementation of the k-means algorithm in a single shot, see below), with plain English.
>
> Moreover, the purpose of this benchmark may be considered as a first step towards conversational no-code / low-code use cases, in which the user can specify a high-level intent in natural language and the system generates a program, which then is revised in a discourse with the model. We realize our first incarnation of this benchmark has certain limitations regarding the “conversational” aspect, but we feel it is a valuable first step towards this goal.
>
> ```python
> # Import libraries.
>
> import numpy as np
>
> # Implement a function to sample n points from a bivariate normal distribution with mean (x_mean, y_mean) and standard deviation (x_std, y_std).
>
> def sample_bivariate_normal(x_mean, y_mean, x_std, y_std, n):
>     x = np.random.normal(x_mean, x_std, n)
>     y = np.random.normal(y_mean, y_std, n)
>     return x, y
>
> # Call the function to sample 100 points named points1 centered at (20, 20) with standard deviation (1, 1).
>
> points1 = sample_bivariate_normal(20, 20, 1, 1, 100)
>
> # Call the function to sample 100 points named points2 centered at (-20, -20) with standard deviation (1, 1).
>
> points2 = sample_bivariate_normal(-20, -20, 1, 1, 100)
>
> # Concatenate these data points.
>
> points = np.concatenate((points1, points2), axis=1)
>
> # Implement the k-means clustering algorithm with n iterations and the centroids as return value.
>
> def k_means(points, k, n):
>     centroids = np.random.uniform(-30, 30, (k, 2))
>     for i in range(n):
>         clusters = np.zeros(points.shape[1])
>         for j in range(points.shape[1]):
>             distances = np.zeros(k)
>             for l in range(k):
>                 distances[l] = np.linalg.norm(points[:, j] - centroids[l, :])
>             clusters[j] = np.argmin(distances)
>         for m in range(k):
>             centroids[m, :] = np.mean(points[:, clusters == m], axis=1)
>     return centroids
>
> # Run the algorithm on the points for 100 iterations with 2 clusters and assign the result to "my_centroids".
>
> my_centroids = k_means(points, 2, 100)
>
> # Assign the centroid with negative coordinates to c1 and the one with positive coordinates to c2.Round the coordinates element-wise to the nearest integers and print the two centroids c1, c2 in the format of "(x1, y1), (x2, y2)".
>
> c1 = my_centroids[0, :]
> c2 = my_centroids[1, :]
> c1 = np.around(c1)
> c2 = np.around(c2)
> print("(%d, %d), (%d, %d)" % (c1[0], c1[1], c2[0], c2[1]))
> ```

---

> ### Author Response · Authors · 2022-11-12
> **Reply from Authors (1/3)**
>
> ### Q1. Evaluation of our models on other datasets.
>
> Thank you for the suggestion! We agree, and evaluated our models on both APPS and MBPP in a one-shot and a zero-shot manner, respectively. For APPS, due to the time constraint, we generated 40 candidate solutions on a subset of the test set consisting of 100 examples. For MBPP, we use the sanitized MBPP (Chen et al., 2022) and sample n=100 programs for each problem. The results are displayed in the table below.
>
> **APPS:**
>
> |                  | pass@5 | pass@40 |
> | ---------------- | ------ | ------- |
> | CodeGen-16B-mono | 3.84   | 10.00   |
> | code-cushman-001 | 7.03   | 11.00   |
> | code-davinci-001 | 19.34  | 44.00   |
> | code-davinci-002 | 35.83  | 56.00   |
>
> We report our preliminary results for the APPS, where 100 Introductory problems (cf. higher level problems yielded zero values across the board and were not representative) are randomly sampled from the test set, and n=40 samples are generated for each problem in an in-context 1-shot manner. This approach corresponds to Chen et al. (2021)’s “1-shot Codex raw Pass@k” metric. CodeGen is reasonably close to code-cushman-001 for pass@40. We will continue this evaluation by running on the full test set with n=1,000 as reported in the previous work.
>
>
> **MBPP:**
>
> |                  | pass@1 | pass@10 | pass@100 |
> | ---------------- | ------ | ------- | -------- |
> | CodeGen-350M-NL          | 0.96   | 6.37    | 19.91    |
> | CodeGen-2B-NL            | 5.34   | 24.63   | 48.95    |
> | CodeGen-6B-NL            | 8.15   | 31.21   | 55.27    |
> | CodeGen-16B-NL           | 10.92  | 38.43   | 62.76    |
> | CodeGen-350M-Multi       | 7.46   | 24.18   | 46.37    |
> | CodeGen-2B-Multi         | 18.06  | 45.80   | 65.34    |
> | CodeGen-6B-Multi         | 18.35  | 47.27   | 67.92    |
> | CodeGen-16B-Multi        | 20.94  | 51.61   | 70.02    |
> | CodeGen-350M-Mono        | 14.59  | 41.49   | 63.00    |
> | CodeGen-2B-Mono          | 27.31  | 59.19   | 74.24    |
> | CodeGen-6B-Mono          | 32.48  | 64.20   | 76.81    |
> | CodeGen-16B-Mono         | 35.28  | 67.32   | 80.09    |
> | InCoder-6B       | 21.30  | 46.50   | 66.20    |
> | code-cushman-001 | 45.90  | 66.90   | 79.90    |
> | code-davinci-001 | 51.80  | 72.80   | 84.10    |
> | code-davinci-002 | 58.10  | 76.70   | 84.50    |
>
>
> Following Chen et al. (2022), we sampled programs from the sanitized MBPP for all of our models, with n=100 and temperature=0.8. The last four rows are from the aforementioned paper. In general, we observe the consistent trend of improving the performance over different versions (NL, Multi, Mono), with our largest CodeGen-16B-Mono approaching the results from code-cushman-001. While we do not know whether any of OpenAI models is the “Codex 12B” reported in Chen et al. (2021), we believe our model achieves reasonable results on MBPP as well. We also note that our CodeGen-6B-Mono significantly outperformed InCoder-6B.
>
> * Chen et al. (2021): Chen, Mark, et al. "Evaluating large language models trained on code." arXiv preprint arXiv:2107.03374 (2021).
> * Chen et al. (2022): Chen, Bei, et al. "Codet: Code generation with generated tests." arXiv preprint arXiv:2207.10397 (2022).

---

### Official Review · Reviewer_9c66 · 2022-10-25

**Confidence:** 4
**Correctness:** 3
**Technical Novelty And Significance:** 2
**Empirical Novelty And Significance:** 4
**Recommendation:** 8

**Clarity, Quality, Novelty And Reproducibility:**

The paper is overall clear and well-written.

Appendix B contains most technical details to make a re-implementation believable, and the source code release supports this assumption.

* (Q1) Will the BigPy dataset be released?
* (Q2) Will MTPB be released?
* (Q3) MTPB: What is the longest intermediate piece of code required in a turn?
* (Q4) MTPB: Did you evaluate how realistic it is to assume that users can provide such close guidance, but not write the code themselves?

Minor comments:
* page 3: "models are called call the models as"
* page 3: "While it is not our focus ..." - this sentence is not grammatical, and I have no idea what point is being made here.
* Fig 1 (nit): the solution produced by the model is wrong (email addresses are much complicated than this), but on par with how most humans would solve the problem.
* Sect. 4.3: this repeats earlier (more complete) descriptions of the used models - this is not necessary.
* page 4: "performance on the MTPB improves as a function of the model size and data size" - the latter claim is unsupported, as the additional data is also more specialized to the problem setting (i.e., more Python-specific). Please rephrase.

**Strength And Weaknesses:**

* (+) Training and releasing the models is in itself forms a substantial contribution to the field, enabling research groups with more limited resources to experiment on top of sound base models of significant sizes.
* (+) The multi-turn dataset is a valuable contribution towards practical neural program synthesis applications.
* (~) MTPB does not allow for "going back" (i.e., a user indicating that a generated response is wrong & clarifying their prompt).
* (-) MTPB (from a cursory qualitative evaluation of examples) seems to rely on very fine-grained prompts. In many examples in App F, the prompts are substantially longer than the required code.


**Summary Of The Paper:**

A family of large decoder-only Transformer models fine-tuned on code data are presented and released for downstream use.
A new "multi-turn programming benchmark" (MTPB) is presented as well, in which models have to generate larger programs step-by-step, with natural language prompts describing each step.
The new models (and some baselines) are evaluated on code generation tasks. First, results on the HumanEval dataset show that the new models are competitive with the state of the art.
Results on the new MTPB dataset again indicate that the newly-trained models are competitive, and show that the new setting allows all models to synthesize more complex programs.

**Summary Of The Review:**

The core contribution here is the release of the CodeGen models, for which alone I'd recommend acceptance of the paper. I find the MTPB dataset to be largely uninteresting, as it relies on very fine-grained natural language supervision. It's not clear to me whether it adds value over the Django dataset of Oda et al.

---

> ### Author Response · Authors · 2022-11-12
> **Reply from Authors (2/2)**
>
> ### Q6. Typos and grammatical errors.
>
> We have corrected those errors and updated the PDF.
>
> ### Q7. Repeated descriptions in Section 4.3.
>
> We have removed the repeated descriptions and updated the PDF.
>
> ### Q8. Rephrase "performance on the MTPB improves as a function of the model size and data size".
>
> The data size indicates the amount of Python data in the training data for the three versions of models: CodeGen-NL, CodeGen-Multi, CodeGen-Mono. The Pile, CodeGen-NL training data, contains a small amount of Python data. The BigCode, CodeGen-Multi training data, contains more Python data. The BigPython, CodeGen-Mono training data, is all Python data. As you pointed out, the additional data is more specialized to the HumanEval benchmark. While it is not surprising, we verified that the performance on HumanEval (based on Python) increases as a function of the amount of Python training data. We clarified this point in the paper.

---

> ### Author Response · Authors · 2022-11-12
> **Reply from Authors (1/2)**
>
> ### Q1. MTPB does not allow for "going back".
>
> Indeed MTPB lacks this property. The main motivation of MTPB is to factorize a complicated problem into multiple simpler subproblems, and to measure a model’s program synthesis capacity in this context. It would be interesting and valuable, as you suggested, to take into account the scenario that users indicate the model’s response is wrong and make clarifications with new prompts. However, this an orthogonal design dimension to our current focus (factorizing a complex problem to simple subproblems).
>
>
> ### Q2. Will the BigPython dataset be released?
>
> Unfortunately, we can not release the exact training dataset as this would require a lengthy clearance process involving several departments. That being said, we provide an open-source implementation of the preprocessing steps to facilitate the reproduction of these datasets. The implementation can be found in our anonymous repository:
>
> https://anonymous.4open.science/r/jaxformer/preprocess/
>
>
> ### Q3. Will MTPB be released?
>
> Yes. The benchmark including data and reference implementation is available here:
>
> 1. Data: https://anonymous.4open.science/r/codegen/mtpb.jsonl
> 2. Sampling: https://anonymous.4open.science/r/codegen/mtpb_sample.py
> 3. Execution: https://anonymous.4open.science/r/codegen/mtpb_exec.py
>
> In addition, the samples used for evaluation of MTPB in the paper can be found here:
>
>
> http://benchmark.codegen-iclr.org/
>
> ### Q4. What is the longest intermediate piece of code required in a turn?
>
> The maximum number of generated tokens is set to 512. Based on our experience, this maximum length is able to cover almost all cases.
>
> ### Q5. Realisticness of the Benchmark. “Did you evaluate how realistic it is to assume that users can provide such close guidance, but not write the code themselves?” “MTPB seems to rely on very fine-grained prompts. In many examples in App F, the prompts are substantially longer than the required code.”
>
> The benchmark involves two types of capabilities from users: (1) general problem-solving capacity, and (2) programming skills. The requirement of (1) is relatively higher than (2) since (1) is required to factorize the complex problems and (2) is only required to judge whether the model responses make sense. Our benchmark is targeted to measure a model’s program synthesis capacity and therefore low requirement on (2) is the focus. While the ultimate goal is to have a model with low requirements on both (1) and (2) for users, this is beyond the scope of the current work. We propose the first multi-turn benchmark as a first step towards such a goal.
>
> We include problems at different difficulty levels. The generated code for easy problems is indeed quite short. The model responses for hard problems are actually quite long. We give one example below.
>
> In particular, the “k-means” problem in MTPB below illustrates complex problems such as sampling from a bi-variate normal distribution or the implementation of an entire algorithm with a single intent.
>
> ```python
> # Import libraries.
>
> import numpy as np
>
> # Implement a function to sample n points from a bivariate normal distribution with mean (x_mean, y_mean) and standard deviation (x_std, y_std).
>
> def sample_bivariate_normal(x_mean, y_mean, x_std, y_std, n):
>     x = np.random.normal(x_mean, x_std, n)
>     y = np.random.normal(y_mean, y_std, n)
>     return x, y
>
> # Call the function to sample 100 points named points1 centered at (20, 20) with standard deviation (1, 1).
>
> points1 = sample_bivariate_normal(20, 20, 1, 1, 100)
>
> # Call the function to sample 100 points named points2 centered at (-20, -20) with standard deviation (1, 1).
>
> points2 = sample_bivariate_normal(-20, -20, 1, 1, 100)
>
> # Concatenate these data points.
>
> points = np.concatenate((points1, points2), axis=1)
>
> # Implement the k-means clustering algorithm with n iterations and the centroids as return value.
>
> def k_means(points, k, n):
>     centroids = np.random.uniform(-30, 30, (k, 2))
>     for i in range(n):
>         clusters = np.zeros(points.shape[1])
>         for j in range(points.shape[1]):
>             distances = np.zeros(k)
>             for l in range(k):
>                 distances[l] = np.linalg.norm(points[:, j] - centroids[l, :])
>             clusters[j] = np.argmin(distances)
>         for m in range(k):
>             centroids[m, :] = np.mean(points[:, clusters == m], axis=1)
>     return centroids
>
> # Run the algorithm on the points for 100 iterations with 2 clusters and assign the result to "my_centroids".
>
> my_centroids = k_means(points, 2, 100)
>
> # Assign the centroid with negative coordinates to c1 and the one with positive coordinates to c2.Round the coordinates element-wise to the nearest integers and print the two centroids c1, c2 in the format of "(x1, y1), (x2, y2)".
>
> c1 = my_centroids[0, :]
> c2 = my_centroids[1, :]
> c1 = np.around(c1)
> c2 = np.around(c2)
> print("(%d, %d), (%d, %d)" % (c1[0], c1[1], c2[0], c2[1]))
> ```

---

> > ### Comment · Reviewer_9c66 · 2022-11-15
> > **Paper Discussion**
> >
> > Thank you for your extensive reply and the updates to the paper. I have two minor follow-up questions:
> >
> > Re your answer to Q4: I assume that you have reference solutions for all examples in MTPB (based on the "max_gen_length" information in the benchmark). Could you produce a plot showing the relationship between the length of the reference solution per turn and the overall pass rate?
> > I'm asking this because in the `k_means` example given in your answer, the "Implement the k-means clustering" turn seems to be much harder than everything else shown in the paper (triple nested loop and all that!). I would be interested in something that groups examples by the longest turn (e.g., int(log(max(tokens_per_turn), 4)) and then shows the pass rate.
> > More generally, I find the categorization into easy/medium/hard difficulity level by pass rates of the models that you are comparing a bit self-referential - I understand that there's no obvious way of categorizing these things here, but this also doesn't seem to be ideal.
> >
> > Re your answer to Q5: My point here is that I have doubts that the skill you are requiring here from the user is "general problem solving capacity." This is based on the prompts given in the dataset, some of which are highly imperative and geared towards a specific implementation, as for example these two:
> >  * Concatenate lst_pos and lst_neg and assign it lst2
> >  * Import the regex module and define a pattern \"pat\" that matches capital alphabets that can be referenced as a group.
> >
> > So my question really is whether it is realistic to assume that someone can provide prompts like this, but is not able to (more quickly) write the equivalent source code.
> > Maybe it would help if you could show the average relationship between #tokens in prompt vs #tokens in reference solution?

---

> > > ### Author Response · Authors · 2022-11-16
> > > **Additional Reply to Q4 and Q5**
> > >
> > > ### Q4.
> > >
> > > Following your suggestion, we provide three plots below illustrating the relation of the maximum length of completions per problem and the corresponding pass-rate:
> > >
> > > - Scatter plot of maximum length of sampled completions for a problem for which all tests have passed (see **Figure 7** in the Appendix)
> > > - Bar chart with grouping following int(log(max(tokens_per_turn), 4)) (see **Figure 8** in the Appendix)
> > > - Bar chart with 15 equi-distant bins over maximum prompt length (see **Figure 9** in the Appendix)
> > >
> > > Note, as we do not have reference solutions per turn, we considered samples from CodeGen-16B-Mono for which all tests passed as reference solutions after manual inspection.
> > >
> > > As you have hypothesized, one may assume that indeed shorter maximum length of samples within a problem yield higher pass rates. It may be a good idea to provide hand-written reference solutions and use the length of those solutions as a proxy for hardness, as you have suggested.
> > >
> > >
> > > ### Q5.
> > > Thank you for the clarification! Indeed, there are cases where writing the source code is faster than writing the prompts, especially for experienced programmers. However, we believe the specification in English is valuable for novice users or learners who are clear about the task but are unfamiliar with how to implement, even if the specifications become lengthy. We show the plot (see **Figure 10** in the Appendix) for token count relationship between prompts and solutions obtained from test-passing CodeGen-16B-Mono samples, as described before. Each point corresponds to one turn pair of (prompt, generated program), and a guide line indicating equal count ratio. On average, the token count ratio: #token_in_program / #token_in_prompt = 1.06.
> > > We would like to note that the model sometimes struggles with a turn and ends up not generating a program for the turn, which leads to zero token count (i.e., points on the x-axis), and therefore #token_in_program underestimates #token_in_reference_solution.

---

### Official Review · Reviewer_wk6b · 2022-10-26

**Confidence:** 4
**Correctness:** 3
**Technical Novelty And Significance:** 3
**Empirical Novelty And Significance:** 3
**Recommendation:** 6

**Clarity, Quality, Novelty And Reproducibility:**

# Clarity
I did not realize until quite late that the $\textsc{CodeGen-Multi}$ and $\textsc{CodeGen-Mono}$ models are trained on, respectively, The Pile plus the BigQuery dataset, and The Pile plus BigQuery plus BigPython. That fact is only mentioned in Section 4.3 and in the appendix, and not where the models are initially described (Section 2.1 and 2.2). This can be clearer in Section 2.

# Originality
The multi-turn dataset introduced in the paper seems to have quite a bit of similarity to SPoC (from Kulal et al., 2019), although SPoC is generally at a lower level of granularity. A more detailed discussion about it would have been helpful.

# Quality
- The full model performs well on the HumanEval dataset, and performance scales with increasing data and model size, showing that the model training worked successfully.
- It is unclear why Table 2 only has information about $\textsc{CodeGen-Mono}$ and not the other models. More data here would be more convincing.
- For the multi-turn dialogue data, the instructions for subsequent turns only make sense if the previous turns generated correct code (or at least code in line with the authors' intent). As such, it would be useful to see how well the model does when all the output for all previous turns are correct.

**Details Of Ethics Concerns:**

For other models trained on code from GitHub and similar sources, like OpenAI Codex, there have been some concerns raised about whether they may violate the copyright of the original authors.

**Strength And Weaknesses:**

# Strengths
- The paper reports state-of-the-art results for the HumanEval benchmark for code generation from natural language specifications.
- The authors provide downloadable checkpoints for the trained models, which can be a useful resource for the community.
- The paper introduces a multi-turn natural language to code dataset where the natural language instructions are interleaved with the intended generated code.

# Weaknesses
- The model is evaluated only on one existing dataset, HumanEval, and not other similar datasets like APPS or MBPP.
- While the work represents a significant expenditure of resources, the amount of technical novelty is relatively low, and some of the new aspects in the construction of the model itself are not studied or described as much as they could be:
  - The \textsc{BigPython} dataset seems responsible for much of the gains in the work, but the description of it in the paper is limited to "We have compiled public, non-personal information from GitHub consisting of permissively licensed Python code in October 2021".
  - There is significant space dedicated to describing the TPU-based training setup with JAX, but not much about the reasoning behind it.
- The paper doesn't have much experiments about hyperparameters and training data selection (e.g. the order in which the datasets were used for training), or further information/justification for the values used.
- There is no release of training datasets.
- The deduplication methodology used during dataset preprocessing may be too weak. Only removing exact duplicate files may still leave many files which are very similar (for example, varying only in some comments at the top of the file). For more discussion on another code dataset, see https://twitter.com/miltos1/status/1497126435261083649
- Considering that copies of the HumanEval dataset may have been posted to GitHub, the authors should investigate the possibility of training dataset contamination.

**Summary Of The Paper:**

The paper presents several large autoregressive language models trained using large datasets scraped from the web, and in particular datasets of code from GitHub. The authors evaluate the models on tasks of generating code from natural language. First, the authors show state-of-the-art results on the HumanEval dataset and report that larger models also demonstrate lower perplexity on the prompt (the natural language program specification). The authors then describe the Multi-Turn Programming Benchmark (MTPB), in which the natural language intents are interleaved with code and each intent describes what the immediately subsequent code should do. Compared to a baseline where all the natural language is provided to the model concatenated together, the multi-turn version shows improvement.

**Summary Of The Review:**

I think the model snapshots from the paper are a highly useful contribution to the community and I based my recommendation largely on that point.

---

> ### Author Response · Authors · 2022-11-12
> **Reply from Authors (2/2)**
>
> ### Q4. Release of the training datasets.
>
> Unfortunately, we can not release the exact training dataset as this would require a lengthy clearance process involving several departments. That being said, we provide an open-source implementation of the preprocessing steps to facilitate the reproduction of these datasets. The implementation can be found in our anonymous repository:
>
> https://anonymous.4open.science/r/jaxformer/preprocess/
>
>
> ### Q5. The deduplication methodology used during dataset preprocessing may be too weak.
>
> Thank you for the informative pointer! We understand that only removing exact duplicates may not be the optimal pruning strategy and would be highly interested to study the effect of methods of deduplication on the model performance. That being said, one could make the handwavy argument that the frequency of observing a file (or near duplicates of such) indicates the performance of code in the corpus (or weakly correlates with the quality of code). That is, frequently used code is more relevant to completions (and may have higher quality) than less frequently code. While we generally agree, it is not entirely clear to us that uniform probability mass across files is optimal.
>
> ### Q6. Contamination of training dataset with HumanEval.
>
> The collection of the dataset concluded in October 2021, while HumanEval was released in July 2021. We may assume that while HumanEval itself may be contained in the dataset, solutions to the benchmark are probably not contained.
>
> However, we will run a full scan to confirm this. Thank you for pointing this out.
>
> ### Q7. Emphasize the training datasets for CodeGen-Multi and CodeGen-Mono.
>
> Thank you for the feedback. We clarified the nature of the sequential training in Section 2.1.
>
> ### Q8. Similarity to SPoC.
>
> The subprogram-level instructions in SPoC indeed resemble the notion of natural language instructions in MTPB. However, we believe, the specifications in SPoC are more akin to traditional pseudo-code than natural language, as we mentioned in our prior art section, while MTPB aims to capture the intent itself in natural language. This may be especially apparent in the provided “k-means” problem, where the task is to implement the entire k-means algorithm given a single prompt.
>
>
> ### Q9. PPL evaluation for other models.
>
> Thank you, we have added perplexity numbers for additional models in Appendix F. Please also see the results below.
>
> Since the HumanEval benchmark is expressed in Python, it was natural to focus on the mono-lingual (Python-only) model, but of course, the additional information is insightful. The purpose of this observation is primarily to motivate the multi-turn factorization of prompts.
>
> | NL   | 350M | 2B   | 6B   |
> | ---- | ---- | ---- | ---- |
> | Pass | 4.53 | 3.25 | 2.78 |
> | Fail | 4.96 | 3.87 | 3.65 |
>
> | Multi | 350M | 2B   | 6B   |
> | ----- | ---- | ---- | ---- |
> | Pass  | 4.78 | 3.82 | 3.82 |
> | Fail  | 5.64 | 4.85 | 4.80 |
>
>
> ### Q10.  it would be useful to see how well the model does when all the output for all previous turns are correct.
>
> Indeed, sampling conditional on previous steps with correct code would be very interesting. Unfortunately, we do not have test cases for step-wise unrolling and thus can not verify the correctness of individual steps, only of the resulting program in its entirety.
>
> ### Q11. Justification of choice of hyper-parameters and selection of training data.
>
> For the hyperparameters, the training hyperparameters in Table 6 primarily follow the reference values of GPT-3 with variations for TPU-specific optimizations. As our compute was limited, we did not have the luxury to run an extensive search over hyper-parameters. We have clarified these considerations in Section A.3.
>
> For the data selection, we have clarified the write-up as you suggested. Once more, we were compute-constrained in terms of exploration. However, we are evaluating various data strategies as we speak.

---

> > ### Comment · Program_Chairs · 2022-11-28
> > **comment from SPC**
> >
> > Authors-please comment on the potential license issue raised by this reviewer.

---

> > > ### Author Response · Authors · 2022-11-29
> > > **Reply from Authors**
> > >
> > > Thank you, we appreciate the reminder to address this concern.
> > >
> > > The ongoing discussion regarding legal compliance (copyright, terms of use, licenses) is a genuine concern and, we believe, remains an open question until there has been a precedent legal case established. This appears an unresolved question, specifically for models trained on code artifacts with associated licenses, and, more generally, for most pre-trained generative models.
> > >
> > > We open-source the model parameters as a means of sampling. The usage of samples drawn from this model requires careful consideration and responsibility from the end users.
> > >
> > > We have revised the “Ethical Consideration” section to raise awareness and make this consideration (and concern) explicit.

---

> ### Author Response · Authors · 2022-11-12
> **Reply from Authors (1/2)**
>
> ### Q1. Evaluation of other benchmarks.
>
> Thank you for the pointers. Following your suggestions, we have evaluated our models on both benchmarks (MBPP and APPS). The results are displayed in the table below.
>
> **MBPP:**
>
> |                  | pass@1 | pass@10 | pass@100 |
> | ---------------- | ------ | ------- | -------- |
> | CodeGen-350M-NL          | 0.96   | 6.37    | 19.91    |
> | CodeGen-2B-NL            | 5.34   | 24.63   | 48.95    |
> | CodeGen-6B-NL            | 8.15   | 31.21   | 55.27    |
> | CodeGen-16B-NL           | 10.92  | 38.43   | 62.76    |
> | CodeGen-350M-Multi       | 7.46   | 24.18   | 46.37    |
> | CodeGen-2B-Multi         | 18.06  | 45.80   | 65.34    |
> | CodeGen-6B-Multi         | 18.35  | 47.27   | 67.92    |
> | CodeGen-16B-Multi        | 20.94  | 51.61   | 70.02    |
> | CodeGen-350M-Mono        | 14.59  | 41.49   | 63.00    |
> | CodeGen-2B-Mono          | 27.31  | 59.19   | 74.24    |
> | CodeGen-6B-Mono          | 32.48  | 64.20   | 76.81    |
> | CodeGen-16B-Mono         | 35.28  | 67.32   | 80.09    |
> | InCoder-6B       | 21.30  | 46.50   | 66.20    |
> | code-cushman-001 | 45.90  | 66.90   | 79.90    |
> | code-davinci-001 | 51.80  | 72.80   | 84.10    |
> | code-davinci-002 | 58.10  | 76.70   | 84.50    |
>
> Following Chen et al. (2022), we sampled programs from the sanitized MBPP for all of our models, with n=100 and temperature=0.8. The last four rows are from the aforementioned paper. In general we observe the consistent trend of improving the performance over different versions (NL, Multi, Mono), with our largest CodeGen-16B-Mono approaching the results from code-cushman-001. While we do not know whether any of OpenAI models is the “Codex 12B” reported in Chen et al. (2021), we believe our model achieves reasonable results on MBPP as well. We also note that our CodeGen-6B-Mono significantly outperformed InCoder-6B.
>
> **APPS:**
>
> |                  | pass@5 | pass@40 |
> | ---------------- | ------ | ------- |
> | CodeGen-16B-mono | 3.84   | 10.00   |
> | code-cushman-001 | 7.03   | 11.00   |
> | code-davinci-001 | 19.34  | 44.00   |
> | code-davinci-002 | 35.83  | 56.00   |
>
> We report our preliminary results for the APPS, where 100 introductory problems (cf. higher level problems yielded zero values across the board and were not representative) are randomly sampled from the test set, and n=40 samples are generated for each problem in an in-context 1-shot manner. This approach corresponds to Chen et al. (2021)’s “1-shot Codex raw Pass@k” metric. CodeGen is reasonably close to code-cushman-001 for pass@40. We will continue this evaluation by running on the full test set with n=1,000 as reported in the previous work.
>
>
> * Chen et al. (2021): Chen, Mark, et al. "Evaluating large language models trained on code." arXiv preprint arXiv:2107.03374 (2021).
> * Chen et al. (2022): Chen, Bei, et al. "Codet: Code generation with generated tests." arXiv preprint arXiv:2207.10397 (2022).
>
> ### Q2. More details on the BigPython dataset.
>
> Regarding the details on the BigPython dataset, more details can be found in Appendix A.1. Specifically, Table 5 contains descriptive statistics. Moreover, pre-processing code for a reproduction of the dataset can be found here:
>
> https://anonymous.4open.science/r/jaxformer/preprocess/
>
> Unfortunately, we can not provide any further information at this point in time without completing a lengthy compliance process from our side, but naturally, we would wish to provide more details.
>
> We hope this situation is understandable.
>
> ### Q3. Reasoning about TPU-based training setup.
>
> Thank you for pointing this out. The details on the training procedure and technical implementation in JAX and TPU have been moved to the Appendix to avoid distracting the reader. The reasoning for this section is to shed light on the technical complexity of training 16B models based on the mechanisms provided in JAX, especially the pjit() operator for SPMD, on which our provided open-source training library JaxFormer is based. At the time of writing the manuscript, there has been no public and open-source reference implementation available to train LLMs on TPU. Our technical implementation, testing, and finally training of the CodeGen models took several months, which resulted in the JaxFormer library.
>
> The purpose of the section in the Appendix is to (1) demonstrate to the community that training state-of-the-art LLMs can indeed be achieved with a compact training library, (2) make TPUs and JAX more accessible to the community, (3) shed light on technical details of how data and model parallelism was implemented for these models. That being said, we will revise the text to clarify new aspects of the construction of the model itself, as you suggested.

---

### Author Response · Authors · 2022-11-12
**Reply to all Reviewers**

Dear Reviewers,

Thank you for your valuable time and insights.

Following your advice, we have

1. Included additional baselines for MTPB such as Codex Cushman 001, Davinci 001, and Davinci 002,
2. Provided a reference implementation for both sampling and execution of MTPB,
3. Included evaluation of our models on additional benchmarks such as APPs and MBPP,
4. Revised the manuscript according to your suggestions.


Once more, thank you for the insightful feedback and we are looking forward to the discussion.

---

### Decision · Program_Chairs · 2023-01-20

**Decision:**

Accept: notable-top-25%

**Justification For Why Not Higher Score:**

There are some minor flaws, which have been partially addressed but not solved during the rebuttal period.

**Justification For Why Not Lower Score:**

This is a sizeable contribution to the community, there is no incorrectness, we should not downplay useful contributions for lack of novelty, so this should get accepted. If wanted, the SAC can lower to Accept (Poster), but I think having this paper as a spotlight would interest enough of the ICLR community.

**Metareview: Summary, Strengths And Weaknesses:**

The paper introduces a family of LMs and LLMs (up to 16B parameters) trained on code generation and to be released. The impact of the release of such backbones to the community should be consequent: they are not the sole existing models for codegen, but they fill a niche between larger (and thus harder to productionize / serve for inference) and smaller (codegen) LMs. The ablations and the dataset cleaning (descriptions) could be improved further. In particular, there should be a study (it is not easy) of training dataset / eval dataset contamination, which the authors acknowledge in the rebuttal. The authors introduce a new benchmark Multi-Turn Programming Benchmark (MTPB) of 115 problem sets, with multi-turn prompts, which seems interesting. It was debated during review and rebuttal and will include strong baselines to compare on. This can be considered as a clear contribution of the paper. The evaluations that are comparable to prior work are adequate but could be improved: only on one existing dataset, HumanEval. Overall, the paper will be a strong addition to the ICLR conference.

**Note From Pc:**

if the above contains the word "oral" or "spotlight" please see: "oral" presentation means -> notable-top-5% and "spotlight" means -> notable-top-25%. As stated in our emails, we are disassociating presentation type from AC recommendations

**Summary Of Ac-Reviewer Meeting:**

N/A